# Intercomparison of long-term ground-based measurements of total, tropospheric and stratospheric ozone at Lauder, New Zealand

**Robin Björklund**[1], **Corinne Vigouroux**[1], **Peter Effertz**[2,3], **Omaira E. García**[4], **Alex Geddes**[5], **James Hannigan**[6], **Koji Miyagawa**[3], **Michael Kotkamp**[5], **Bavo Langerock**[1], **Gerald Nedoluha**[7], **Ivan Ortega**[6], **Irina Petropavlovskikh**[2,3], **Deniz Poyraz**[8], **Richard Querel**[5], **John Robinson**[5], **Hisako Shiona**[5], **Dan Smale**[5], **Penny Smale**[5], **Roeland Van Malderen**[8], and **Martine De Mazière**[1]

[1]Department of Atmospheric Composition, Belgian Institute for Space Aeronomy (BIRA-IASB), Brussels, Belgium
[2]Cooperative Institute for Research in Environmental Science, University of Colorado- Boulder, Boulder,CO
[3]NOAA Global Monitoring Lab, Boulder, CO
[4]Izaña Atmospheric Research Center (IARC), State Meteorological Agency of Spain (AEMET), Santa Cruz de Tenerife, Spain
[5]National Institute of Water and Atmospheric Research Ltd (NIWA), New Zealand
[6]Atmospheric Chemistry, Observations & Modeling, National Center for Atmospheric Research, Boulder, CO, USA
[7]Naval Research Laboratory, Washington, DC, USA
[8]Royal Meteorological Institute of Belgium, Ringlaan 3, 1180 Uccle (Brussels), Belgium

**Correspondence:** R. Björklund (robin.bjorklund@aeronomie.be)

**Abstract.** Long-term, 21[st]-Century ground-based ozone measurements are crucial to study the recovery of stratospheric ozone as well as the trends of tropospheric ozone. This study is performed in the context of the LOTUS (Long-term Ozone Trends and Uncertainties in the Stratosphere) and TOAR-II (Tropospheric Ozone Assessment Report, phase II) initiatives. Within LOTUS, we want to know why different trends have been observed by different ground-based measurements at Lauder. In TOAR-II, intercomparison studies among the different ground-based data sets are needed to evaluate their quality and relevance for trend studies. To achieve these goals, we perform an intercomparison study of total column ozone and its vertical distribution among the ground-based measurements available at the Lauder station from 2000 to 2022, which are a Fourier transform infrared (FTIR) spectrometer, a Dobson spectrophotometer, a UV2 (Ultra-Violet double monochromator), a microwave radiometer (MWR), ozonesondes, and a stratospheric lidar. Because only the two latter techniques provide high-vertical resolution profiles, the vertical ozone distribution is validated using partial columns, defined to provide independent information: one tropospheric and three stratospheric columns. Because FTIR provides total columns and vertical information covering all partial columns as well as high temporal sampling, the intercomparisons (bias, scatter, and drift) are analyzed using FTIR as the reference.

A very good agreement between the FTIR and Dobson (FTIR and UV2) total column ozone records is apparent in the high Pearson correlation of 0.97 (0.93), low biases of -3% (-2%), and the 2% (3%) dispersions which are within the respective systematic and random uncertainties. The small observed drifts 0.4 (0.3) %/decade are 'non-significant' (or rather a low certainty in a 95% confidence interval) and show good stability of the three ozone total column series at Lauder.

In the troposphere we find a small bias of -1.9% with the ozonesondes, but a larger one (+10.7%) with Umkehr which can be explained by the low degrees of freedom for signal (0.5) of Umkehr in the troposphere. However, no significant drift is found between the three instruments in the troposphere, which proves their relevance for trend studies within TOAR-II. The negative bias observed in total columns is confirmed by negative biases in all stratospheric columns for all instruments with respect to FTIR (between -1.2% and -6.8%). This, confirmed by the total column biases, points to a 2-3% underestimation of the infrared spectroscopic line intensities. Nevertheless, the dispersion between FTIR and all techniques is typically within 5% for the stratospheric partial columns, in close agreement with the given random uncer-

tainty budgets.

We observe no significant drift in the stratosphere between ozonesondes and FTIR for all partial columns, with ozonesonde trends being less negative than in LOTUS (Godin-Beekmann et al., 2022, further referred to as the LOTUS22). The only significant drift in the lower stratospheric columns is obtained between FTIR and Umkehr, as was already found in LOTUS22. Two significant positive drifts are observed in the middle stratosphere (2 and 3 %/decade) with lidar and MWR, respectively, while two significant negative drifts are observed in the upper stratosphere (-3 and -4 %/decade), with Umkehr and lidar, respectively. While remaining drifts are still present, our study explains roughly half of the differences in observed trends in LOTUS22 by the different sampling, vertical sensitivity or time periods and gaps. In addition, the FTIR data in the present work has been improved since LOTUS22, reducing the differences in the upper stratospheric and tropospheric trends. This shows the necessity for continuous review and improvement of the measurement and retrieval processes. This study also reflects the importance of super sites such as Lauder for cross-validating the long-term ozone measurements. Our study demonstrated that well harmonized, optimized, well characterized instruments that show very good agreement in terms of bias, dispersion, and correlation are capable of detecting trends that agree within their respective measurement uncertainties.

## 1   Introduction

The study of ozone plays a crucial role in understanding the effects of climate change, as well as its impact on human health and plant life (see e.g., WMO, 2022; Brasseur and Solomon, 2005). It is common to distinguish between stratospheric ozone and tropospheric ozone. Stratospheric ozone, commonly referred to as the 'ozone layer', serves as our primary defense against harmful UV radiation. The discovery of the hole in the ozone layer, which is most prominent over Antarctica in the austral spring but also present in the Arctic (Solomon, 1999; Manney et al., 2011), emphasized the need for monitoring and reducing the depletion of ozone. The Montreal protocol of 1987 was implemented to reduce the emissions of ozone-depleting substances (ODSs), which are known catalysts for the chemical destruction of ozone. Over the past 30+ years stratospheric ozone has been closely monitored through ground-based, in-situ, and satellite observations. As a result of the decreased stratospheric chlorine levels (Jones et al., 2011), the depletion of the ozone hole has been halted and there are indications of its recovery (Solomon et al., 2016). This recovery is particularly evident in the Antarctic, while in regions between 60°S-60°N, the increase in ozone may be offset by a continued decrease in the lower stratosphere (Ball et al., 2018). Similarly, in the Arctic it is currently unclear if there are any positive ozone trends since 2000 due to the larger dynamical variability complicating the observation of this ozone recovery (WMO, 2018).

Tropospheric ozone is a greenhouse gas that contributes to global warming (Hansen et al., 1997) and poses a significant threat to human health through its effects on the respiratory system (see e.g., Kim et al., 2020). Unlike stratospheric ozone, tropospheric ozone has a relatively short atmospheric lifetime of hours to weeks (Stevenson et al., 2006). It does not have any direct emission sources; rather, it is a secondary gas formed by the interaction of sunlight with hydrocarbons and nitrogen oxides, which are emitted by various human-made sources such as vehicles, fossil fuel power plants, and industrial activities (Jacob, 2000). The combination of surface-ozone, ozonesondes, and aircraft measurements, with satellite measurements provides a long-term collection of data to study tropospheric ozone trends. These measurements show an increase of ozone since the 1990s in the tropics and in northern mid-latitudes (Gulev et al., 2021; Szopa et al., 2021).

This study is done within the context of LOTUS (Longterm Ozone Trends and Uncertainties in the Stratosphere). The current goal of LOTUS is to gain a comprehensive understanding of ozone trends, including their relationship to altitude and latitude, by thoroughly evaluating uncertainties in trend studies and considering the impact of errors related to the sampling and stability of data sets. Within LOTUS, Godin-Beekmann et al. (2022) use a regression model to obtain trends of the stratospheric ozone vertical distribution. They find significant differences in trends between the measurements at Lauder, New Zealand, which is a station in the Network for the Detection of Atmospheric Composition Change (NDACC, De Mazière et al., 2018; Kurylo and Solomon, 1990). Comparisons between these instruments at Lauder have been performed such as by McDermid et al. (1998) who look at several ozone profilers (lidar, microwave radiometer, and ozonesonde). These comparisons have been done before 2000, however, so to analyze the recent differences between the ground-based measurements we look at comparisons done since 2000, such as by Bernet et al. (2020). They only use two of the ground-based measurements (stratospheric lidar and microwave radiometer MWR) together with the Aura Microwave Limb Sounding satellite and ERA5 reanalysis data, and thus do not explain the differences for most measurements in Godin-Beekmann et al. (2022). Similarly Steinbrecht et al. (2017) use the microwave radiometer, lidar, FTIR (Fourier transform infrared) spectroscopy, and Dobson Umkehr measurements in combination with satellite measurements to compute trends representative for the whole southern latitude band (35-60°S). There are some differences for FTIR and lidar in the lower stratosphere, but they find similar statistically significant values in the upper stratosphere where they calculate a trend of 2.5 %/decade and thus strengthen those results from Godin-Beekmann et al. (2022).

We will look at all of the ground-based measurements which were discrepant in Godin-Beekmann et al. (2022), namely the FTIR spectrometer, Umkehr measurements from the Dobson spectrophotometer, ozonesondes, and the stratospheric lidar. We will add the MWR as in Bernet et al. (2020), which was not included in Godin-Beekmann et al. (2022) because it stopped in 2016. We will also add to our analysis total column intercomparisons from the FTIR, the Dobson spectrophotometer, and a UV2 (Ultra-Violet double monochromator). In our intercomparison study we aim to quantify the biases and potential drifts between the different measurements to analyze whether the discrepancies observed in the stratospheric trends at Lauder (Godin-Beekmann et al., 2022) could be due to the different sampling or vertical resolution of the measurements.

Tropospheric ozone intercomparisons will also be performed using FTIR, ozonesondes and Umkehr data sets. This tropospheric work is made within the context of the HEGIFTOM focus working group (Harmonization and Evaluation of Ground Based Instruments for Free Tropospheric Ozone Measurements, http://hegiftom.meteo.be) within the TOAR-II (Tropospheric Ozone Assessment Report phase II) initiative. This working group focuses in part on the need for a thorough intercomparison of tropospheric ozone measurements, where all biases and drifts of the used instruments are evaluated. This study will provide a detailed look on these biases and drifts which will serve as a reference to future TOAR-II studies using these data sets.

First, in Section 2, we elaborate on the different ground-based ozone measurements. Next, in Section 3, we explain the intercomparison method where we select coincident measurements based on the sampling and adjust for differences in vertical resolution. Because we do not have high vertical resolution for all measurements, we divide the vertical ozone profile into four different partial columns, one in the troposphere (0-5-11 km) and three in the stratosphere (14-22 km, 22-29 km, and 29-42 km). Because FTIR provides total columns and vertical information covering all partial columns, as well as a good temporal sampling, the intercomparisons are performed using the FTIR as the reference. In Section 4 we we discuss the obtained total and partial column biases, scatters and drifts of all available ground-based techniques with respect to FTIR, putting them in perspective with the uncertainty budgets. We end with a summary and conclusions in Section 6.

## 2   Ground-based measurements

Five ground-based measurement techniques which have vertical information, available at the Lauder station (45°S, 170°E) are considered in this study: the Fourier transform infrared spectrometer (FTIR), the Dobson Umkehr method (Umkehr), the microwave radiometer (MWR), the stratospheric ozone lidar (lidar), and the ozonesonde observations (Sonde). Some of the specifications for each of these measurements are given in Table 1. Aside from these five profile measurements, we consider two total column ozone (TCO) measurements: Dobson TCO (Dobson), which is separate from the Dobson Umkehr technique, and Bentham ultraviolet double monochromator (UV2, Geddes et al., 2024). The FTIR and Umkehr techniques also provide total columns. In Table 1 we identify a group of low vertical resolution profiles (FTIR, MWR, and Umkehr) where profiles are derived from an inversion method by Rodgers (2000), and for which we give the degrees of freedom for signal. The second group contains vertically resolved measurements, which includes the Sonde and lidar. The time series of the total ozone column for all these measurements are shown in Figure 1. Note that for Sonde, MWR, and lidar we show the integrated column of the available data which covers only a part of the total ozone profile as we can see in Table 1, which is why the absolute value is shifted downward. The figure shows us the coverage of the observations over the full time span. This displays a more densely sampled time series for the FTIR, Umkehr, MWR, and UV2 measurements as opposed to the less frequent observations of lidar and the ozonesonde. Additionally, we see a shorter time span for the MWR data (which stops in October 2016) and for UV2 (which starts in 2012) and a gap of the lidar data between 2012 and 2015. We consider observations made after 2000 which is the starting year used in recent studies (Godin-Beekmann et al., 2022; WMO, 2022) studying the stratospheric ozone recovery expected by the reduction of ODSs.

### 2.1   FTIR measurements

The FTIR instruments record mid-infrared solar transmission spectra at high spectral resolution. The spectra contain the signatures of molecular rotational-vibrational transitions of numerous trace gases (including ozone) in the terrestrial atmosphere as they absorb solar radiation. The spectra are analyzed to measure the total columns as well as vertical concentration of these trace gases in the atmosphere using the pressure and temperature dependence of the absorption line shapes. For ozone, the FTIR vertical sensitivity is good from the surface to approximately 50 km (Vigouroux et al., 2008). The retrieval strategy is based on the inversion method developed by Rodgers (2000) and the details are described in Vigouroux et al. (2008). An important aspect of the method is the necessity of a priori vertical profiles of both the target gas and interfering species (those species that have absorption lines in the same spectral window as is considered for the analysis of the target gas) as well as the a priori covariance matrices of the target gas (and interfering species). The vertical information is characterized by the averaging kernel $\mathbf{A}$, defined through

$$\mathbf{x} = \mathbf{x}_a + \mathbf{A}\left(\hat{\mathbf{x}} - \mathbf{x}_a\right) + \boldsymbol{\epsilon}_x, \qquad (1)$$

**Table 1.** Specifications for the ozone measurements available at the Lauder station. These are average, or indicative values for the measurement frequency, altitude boundaries and degrees of freedom for signal (DOFS, see text for definition) if relevant (otherwise Not Applicable, NA), whether or not they measure the total column ozone (TCO) and time period.

|          | Observing frequency | Vertical extent       | DOFS | TCO | Time series          |
| -------- | ------------------- | --------------------- | ---- | --- | -------------------- |
| FTIR     | ~5 per week         | Surface to ± 50 km    | ~4.5 | yes | 2001-2022            |
| Umkehr   | 4-5 per week        | Surface to ± 50 km    | ~3.5 | yes | 1987-2020            |
| MWR      | 2-3 per day         | ± 20 to 50 km         | ~7   | no  | 1992–2016            |
| Lidar    | ~1 per week         | ±10 to 50 km          | NA   | no  | 1994-2011, 2015-2021 |
| Sonde    | ~3 per month        | Surface to ± 30 km    | NA   | no  | 1986-2022            |
| Dobson   | ~1 per day          | NA                    | NA   | yes | 1987–2020            |
| UV2      | ~2 per day          | NA                    | NA   | yes | 2012-2022            |

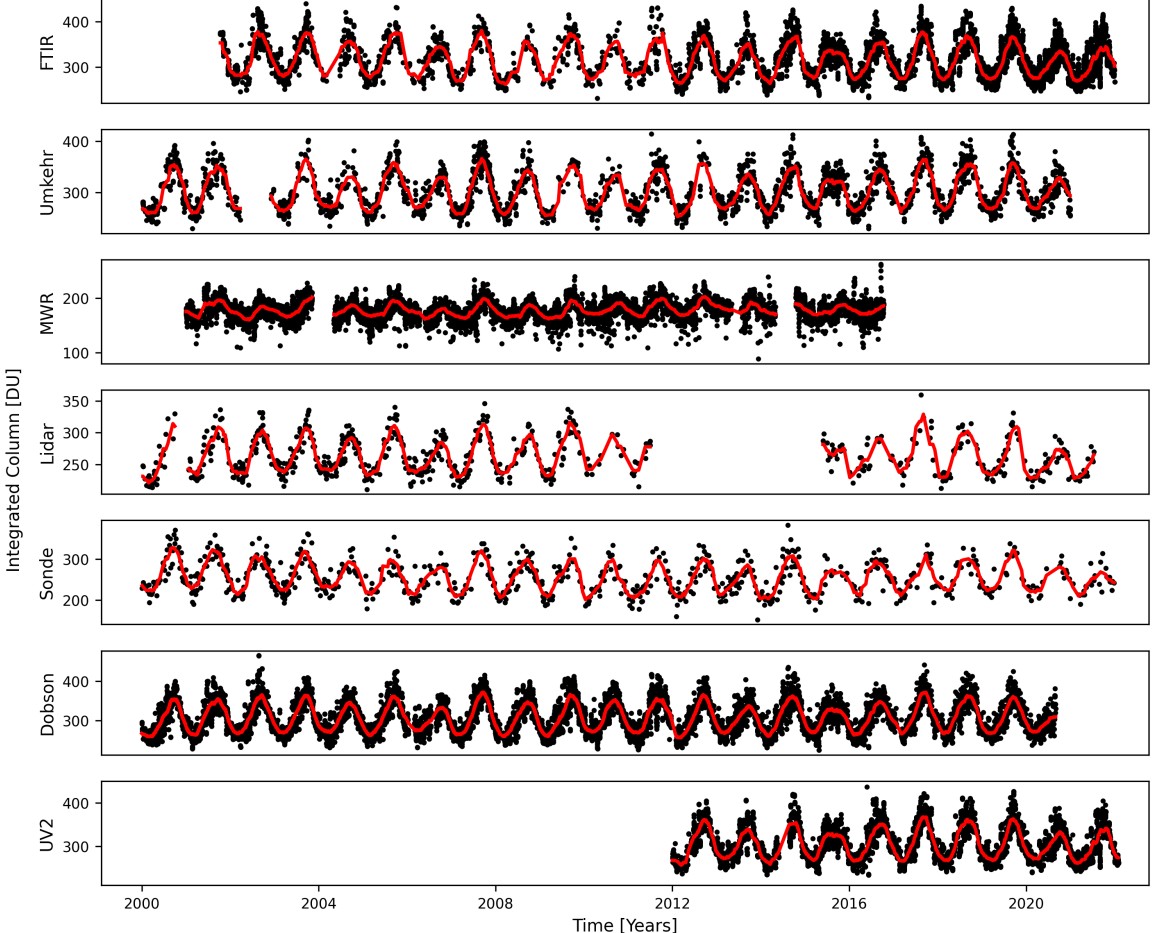

**Figure 1.** Time series since 2000 of the total or integrated ozone column in Dobson Units (DU) for FTIR, Umkehr, MWR, lidar, Sonde, Dobson, and UV2 (from top to bottom). The red line shows a running mean of the data with a window of three months. The integrated column for MWR, lidar, and sonde does not equal full total column of ozone because their limited range of measurement in altitude.

which relates the retrieved state vector $\mathbf{x}$ to the true state $\hat{\mathbf{x}}$ and the a priori state vector $\mathbf{x}_a$, where $\boldsymbol{\epsilon}_x$ contains the measurement and forward model errors. As an example, Figure 2 shows the averaging kernel for one measurement relative to the a priori. Additionally, the figure shows the sensitivity

of the retrieval, which is the sum of the rows of the averaging kernel and indicates the contribution of the measurement to the retrieval compared to the contribution of the a priori. Zero sensitivity would mean that only the a priori contributes to the retrieved profile and the retrieved profile is independent

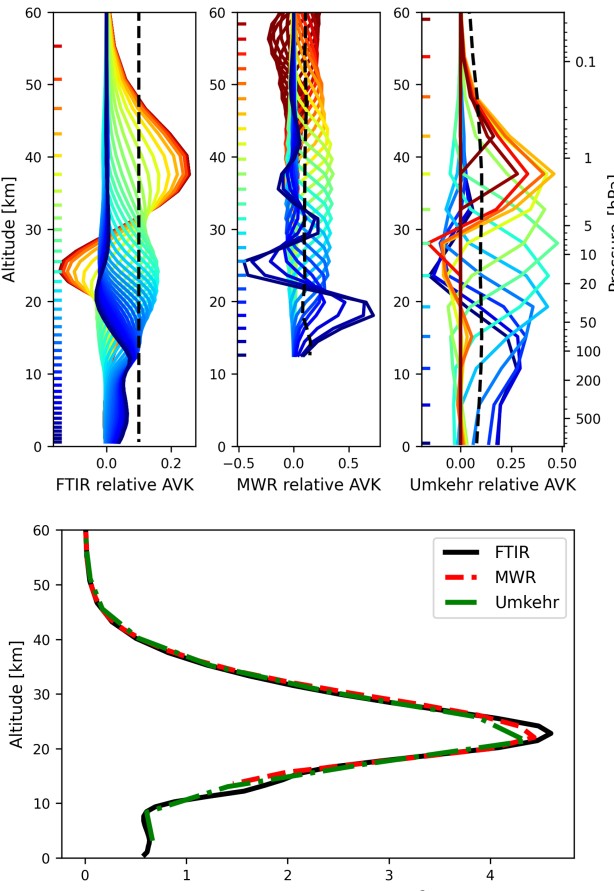

**Figure 2.** Top figure: Averaging kernel of the three low-resolution instruments. The color corresponds to the altitude marked on the left-hand side of the figures. The black-dashed line shows the sensitivity (divided by 10 to better fit on the scale of the figure). Bottom figure: A priori profile of ozone for FTIR, MWR, and Umkehr. The FTIR retrievals use one fixed prior, while both MWR and Umkehr have priors changing in time. For the latter two, we show a mean profile over all measurements within the considered time frame.

from the real profile. These a priori profiles are additionally shown in Figure 2, which, for FTIR, is the same for every retrieval irrespective of when the spectrum is taken.

The selection of lines from the full solar spectrum (also called microwindows), the a priori information, spectroscopic database, etc., all influence the retrieved ozone profile and thus their treatment requires great care and have been harmonized within the InfraRed Working Group (IRWG, www2.acom.ucar.edu/irwg) of the international collaboration NDACC (Vigouroux et al., 2008, 2015). The amount of vertical information that can be obtained is quantified by the degrees of freedom for signal (DOFS) being the trace of the averaging kernel matrix. The total DOFS depend on this retrieval strategy, but of course also on which gas is retrieved and the instrument with which it is observed. For the ozone

measurements at Lauder, the total DOFS are typically around 4 to 4.5.

Since the ozone retrieval strategy plays a crucial rule in the information that can be obtained from the spectra, we use new retrieval strategy throughout this work, which is explained in more detail in Appendix A. Whenever we use the older FTIR strategy as laid out in Vigouroux et al. (2008), for example to compare to previous studies, this will explicitly be stated using 'FTIR V08' to avoid confusion.

For each of the measurements, we need to consider the instrument uncertainties during the intercomparison study (see Table 2). These are divided into systematic uncertainties and random uncertainties, which capture the accuracy and the precision of the measurement respectively. The sources of the FTIR random uncertainties are the measurement noise, errors in the forward model parameters, and the smoothing error due to the low vertical resolution. This smoothing error, however, is considered separate from the other errors as suggested by the NDACC IRWG. For FTIR, the biggest contribution, apart from the smoothing error, to the random uncertainty is typically from the temperature profile (Vigouroux et al., 2008; García et al., 2012). In the case of the systematic uncertainty, the largest contribution actually comes from the spectroscopic database. Specifically, the ozone line strength parameter has the biggest uncertainty in the FTIR retrievals. We set it to be 3% (Gordon et al., 2022). While the ozone infrared line intensities have been improved to match better the UV cross-sections, with an accuracy of 1% for the more intense lines, the systematic uncertainties for the weaker lines are still set to 2-5% (Gordon et al., 2022). The FTIR choice of microwindows includes both intense and weak lines, with different dependence in pressure and temperature, which ensures to achieve the desired vertical information (4-5 DOFS). As explained in more detail in García et al. (2012), the assumed uncertainty sources are propagated to an uncertainty on the retrieved profile following the technique of Rodgers (2000), which is captured by a full error covariance matrix $\mathbf{S_x}$. Additionally, the smoothing error is calculated from the averaging kernel and the a priori covariance matrix $\mathbf{S_a}$ through

$$U_{\mathrm{smooth}} = (\mathbf{A} - \mathbf{I})\,\mathbf{S_a}\,(\mathbf{A} - \mathbf{I})^{T}, \qquad (2)$$

with $\mathbf{I}$ the unit matrix, and $\mathbf{S_a}$ is obtained from the Whole Atmosphere Community Climate Model (WACCM, Marsh et al., 2013) as an estimation for the error covariance. With this information we can assess which errors dominate at which altitudes, which we have calculated for one year worth of data at Lauder. We see that the smoothing error always has the largest contribution in the troposphere. Concerning the other random errors, the uncertainty on the temperature profile has the second biggest contribution followed by the measurement noise. Above 25 km, the random uncertainty is usually dominated by the temperature profile. As for the contributions to the systematic uncertainty, as mentioned above,

the largest contribution comes from the spectroscopic parameters; most notably the line strength parameter (around 3%) followed by the spectroscopic pressure and temperature broadening parameters respectively (around 1-2%). The uncertainty on the temperature profile also plays a role in the contribution to the total systematic uncertainty, starting to become more important above 25 km.

## 2.2 Microwave radiometer

The microwave ozone radiometer at Lauder measures the spectrum produced by a thermally excited rotational ozone transition at 110.836 GHz. Similar to the FTIR retrieval method described above, the profile retrieval makes use of the change in pressure broadening as a function of altitude with a vertical extent between approximately 20 and 50 km. The DOFS here, also computed from the trace of the averaging kernel matrix, are higher than those of FTIR at around 7. An example of the MWR averaging kernel is shown in Figure 2. The experimental technique was described in Parrish et al. (1992), and technical details on the instrument used for this work are given in Parrish (1994).

A formal error analysis for the microwave ozone measurements was presented in Connor et al. (1995), and error estimates for Lauder were updated in Tsou et al. (2000). The net precision was determined to be 5-7% between 56 and 1.3 hPa, and the accuracy 7-9%. The vertical resolution of the Lauder measurements is ∼6-8 km (FWHM, full-width at half-maximum) near 10 hPa. Vertical profiles are provided up to 68 km, however retrievals in the mesosphere have somewhat degraded accuracy, precision, and vertical resolution relative to those in the stratosphere.

## 2.3 Dobson Umkehr and TCO

The Umkehr method of estimating the vertical profile of ozone using the Dobson ozone spectrophotometer has been performed since the early 1930s (Walshaw, 1989; Gotz et al., 1934) and the Umkehr measurements stand as some of the longest vertical profile records of ozone collected. The Umkehr method is performed using the 'C' wavelength pair of the Dobson ozone spectrophotometer taken during zenith sky measurements at 12 nominal solar zenith angles (SZA) across the range of 60-90°. A prism inside the Dobson breaks down the sunlight spectra and a pair of slits rejects light outside a narrow band of chosen wavelengths. The 'C' wavelength pair consists of a 'short' wavelength (centered on 311.5 nm) and a 'long' wavelength (centered on 332.4 nm). In this range of wavelengths of UV light, absorption by ozone decreases with wavelength creating differential observations between strong and non-absorbing solar spectrum. The log of the ratio of measured intensities is called "N-value". The N-value changes between 90 and 60° SZA as the sun rises or sets and the instrument records the so-called "Umkehr"

curve that is sensitive to the vertical ozone distribution above the instrument.

The current NOAA operational Umkehr retrieval algorithm (Petropavlovskikh et al., 2005) uses the Bass and Paur (1985) absorption cross section with its temperature dependence. The profiles are derived using optimal estimation technique (Rodgers, 2000) that includes the a priori ozone profile from ozone climatology (McPeters and Labow, 2012), measurement error and a priori uncertainties which defines the averaging kernels and vertical resolution of retrieved profiles (see Figure 2 for an example of averaging kernels for Umkehr). The measurements allow vertical information to be retrieved from the surface to approximately 50 km. The vertical information that we can get from the averaging kernel matrix for the typical Umkehr measurements give DOFS of around 3.5.

The Lauder Umkehr record is homogenized by the method described in Petropavlovskikh et al. (2022) to create a coherent record for the long-term trend analysis of monthly mean anomalies. The method includes a correction to the Umkehr profiles for the stray light that is outside of the nominal bandpass of the Dobson instrument but is not completely filtered out by slits. The record is homogenized using the MERRA-2 GMI (Modern-Era Retrospective Analysis for Research Applications version 2 Global Modeling Initiative) model as a reference to minimize step changes in the record caused by a change in the optical characteristics of the instrument. Individual N-values have a correction factor at each SZA as described in Petropavlovskikh et al. (2022).

A full uncertainty analysis of the Umkehr observations at Lauder is currently not available, but we can estimate the uncertainties from similar measurements at Boulder, USA, using the same technique (see Table 2 for the random and systematic uncertainties used in this study). This provides us with an analysis of the measurement noise as well as the smoothing error in function of altitude. This measurement noise is calculated from simulated MERRA-2 GMI profiles with added random noise based on the error matrix of Umkehr observations. Considering that the measurement noise is only part of the random error in the remote observations, this estimated uncertainty is an underestimation of the true random error. The smoothing error is calculated using Equation (2) using $S_a$ from the climatology described in Kramarova et al. (2013).

Umkehr total column ozone (from now dubbed Umkehr TCO) is integrated from the vertical profiles supplied by the Umkehr retrieval algorithm. The retrieved integrated Umkehr TCO is constrained by the algorithm to be within the measurement uncertainty of the observed TCO from the Dobson measurements. Additionally, regular TCO measurements are made with the Dobson ozone spectrophotometer (from now dubbed Dobson TCO) using the log of the ratio of solar irradiances made during either direct sun and zenith sky measurements made at regular nominal times (usually one near local solar noon, one in the morning and one in the

afternoon). Dobson TCO is derived typically from the A (305.5/325 nm) and D (317.5/339.9 nm) or C and D wavelength pairs with corrections made for Rayleigh scattering and airmass (Evans and Komhyr, 2008). The use of double pairs is designed to minimize the influence from aerosol scattering and clouds. The precision of the Dobson measurement from direct sun observations is around 1% with an accuracy estimated to be around 5% (Basher, 1985). It can be seen in Figure 1 that the sampling of the Umkehr is reduced compared to direct Dobson TCO measurements. Therefore, for the total column comparisons with FTIR, the Dobson TCO will be preferred to the Umkehr integrated columns.

## 2.4 Lidar

The Lauder stratospheric ozone lidar is a differential absorption lidar (DIAL) that was installed by RIVM (Rijksinstituut voor Volksgezondheid en Milieu) and began measurements in 1994. This lidar system and first results are described in Swart et al. (1994). It relies on two different beams to extract information about the vertical ozone distribution from approximately 10 to 50 km in altitude. It used an XeCl excimer laser for the primary 308 nm beam and a 2-metre Raman cell of Hydrogen gas at 1.3 Bar to produce a 353 nm beam. The 308 nm light is moderately absorbed by ozone while the 353 nm light acts as the reference beam. The time-of-flight from pulse emission through to detection gives the vertical distribution information and the abundance is computed from the ratio of 308 nm and 353 nm backscattered light signals. Over the instrument's multi-decadal lifetime, a number of intercomparisons have been staged with the NDACC travelling standard ozone lidar (e.g., Keckhut et al., 2004; Bernet et al., 2020). Currently, measurements are available up to July 2021, when the excimer laser failed. The lidar measurements are well resolved in altitude with the resolution standardized within NDACC according to Leblanc et al. (2016). This resolution ranges from a few hundred meter at 10 km to several kilometers in the upper stratosphere at 50 km. Measurements are expected to resume during the 2023 calendar year. Concerning the lidar uncertainties (see Table 2), there is a systematic uncertainty of approximately 3%, which is mainly associated with uncertainty in the determination of the temperature-dependent absorption cross-section differential. The assumed random uncertainty for lidar, on the other hand, comes from the photon-counting noise.

## 2.5 Ozonesonde

Ozonesondes are small balloon-borne instruments carried by weather balloons and attached to radiosondes to measure the vertical ozone distribution from the surface to altitudes between 30 and 35 km. The ozonesondes launched in Lauder since 1986 are electrochemical concentration cell (ECC) ozonesondes. In these ozonesondes, a small gas sampling pump forces the ambient air through the cells that are filled with a neutral-buffered potassium iodide sensing solution. The principle of the ozone measurement is then based on the titration of ozone in this solution, so that each $O_3$ molecule causes (ideally) two electrons to flow in the external circuit. This measured electrical current can then be converted to ozone partial pressure, by knowing the gas volume flow rate of and the temperature in the pump. At Lauder, ozonesondes from the two different ECC ozonesonde manufacturers (SPC and EnSci, switch made in May 1994) have been launched, and different sensing solution types (SST1.0 and SST0.5, changed in August 1996) been used as well. As biases exist between those different manufacturer ozonesonde types and between different sensing solution types, the Lauder ozonesonde time series had to be corrected for such biases. Additionally, the location of the pump temperature sensor changed throughout the years, which should also be homogenized. These additional processing steps have been taking into account in the reprocessed, homogenized, Lauder ozonesonde time series here, following the Ozonesonde Data Quality Assessment guidelines of Smit and Oltmans (2012). More details of the Lauder ozonesonde time series and the homogenization procedure can be found in Fig.1, Table 1, and Appendix A of Zeng et al. (2024). The quoted overall precision and uncertainty of onzonesonde measurements are 3-5% and 5-10% respectively (Smit et al., 2021, see WMO/). The systematic uncertainties for the ozonesonde measurements in this study are claimed to be removed thanks to the homogenization process. The random uncertainty on the measured ozone partial columns for the ozonesonde are shown in Table 2. A mean value of the uncertainties on the profile is taken to represent the uncertainty on the partial columns, which is likely an overestimation of the actual uncertainty giving values between 4 and 7%.

In a worldwide ozonesonde comparison with satellite and ground-based total column ozone and with satellite stratospheric O3 profiles, Stauffer et al. (2022) mentioned a negative ozone bias in the homogenized Lauder ozonesonde time records in more recent years. This feature might be related to the so-called "post-2013 dropoff in total ozone" identified in a number of ozonesonde stations (not Lauder) in Stauffer et al. (2022), but, as no clear cause has been determined yet, no correction strategy has been implemented in the here applied WMO/GAW 2021 homogenization procedures. While Stauffer et al. (2022) do not identify Lauder as a dropoff site, in this study we will discuss the implications of a potential 3% post-2016 dropoff in TCO to the trend and drift values of the ozonesonde measurements. Our TCO measurements for the ozonesondes are shown in Figure 1 where a post-2016 dropoff could be present, but it is better seen in this comparison to satellites on the plot in https://acd-ext.gsfc.nasa.gov/anonftp/acd/shadoz/nletter/stations_vs_satellites_timeseries.zip.

## 2.6   UV2

Another TCO measurement available at Lauder is UV2 (Geddes et al., 2024), which is based on a Bentham UV spectrometer. UV2 makes alternating measurement of UV global and direct-sun irradiance, the direct sun spectra are then combined with Dobson-like slit functions and used to calculate ozone using the Dobson method. In Geddes et al. (2024), UV2 ozone was shown to closely agree with Dobson TCO, with a mean bias between them of 2.57 DU and a standard deviation of 1.15 DU. The precision of UV2 is therefore assumed to be consistent with the Dobson TCO at 1%, the accuracy, determined by the mean daily standard deviation is also calculated to be approximately 5%. UV2 observations are available from 2012-2022 and throughout the day.

## 3   Intercomparison setup

### 3.1   Partial column definition

All measurements do not share the same observation time, vertical extent, or vertical resolution. In order to perform a meaningful intercomparison between the various instruments, a consistent validation setup is necessary. For example, because the DOFS of several instruments are low, we do not have well resolved information of the ozone concentration with altitude. Therefore, the vertical range is subdivided into altitude layers, where we measure 'partial columns' of ozone. Similar to the approach in Vigouroux et al. (2008), these partial columns are defined by looking at the vertical information of the reference FTIR measurements as well as the altitude boundaries every other instrument measures at. We disregard profiles from MWR, lidar, or ozonesonde if they stop in the middle of a partial column such that no discontinuity needs to be accounted for in the intercomparison. Ideally, the partial columns should have approximately 1 DOFS to ensure that the retrieved information comes mainly from the measurement and not from the a priori partial columns, meaning that for a total of ~4.5 DOFS for the FTIR profile, we can define 4 altitude layers. The first layer is defined from the surface (~0.5 km) to 11 km (corresponding to the mean tropopause height at Lauder, Sakai et al. (2016)), the second from 14 km (the onset of most LIDAR observations) to 22 km, the third from 22 km to 29 km, and the last from 29 km to 42 km. This upper limit is chosen such that the partial column covers all lidar observations and no extrapolation is needed, which would be the case if the upper limit would be chosen higher. This way, all FTIR observations have around 1 DOFS per partial column. From the ozone profiles we then integrate the partial columns, ending up with one tropospheric column and three stratospheric columns (from now on labelled 'lower', 'middle', and 'upper' stratosphere). In the troposphere we can compare FTIR, Umkehr and ozonesonde measurements; in the lower strato-

sphere we additionally have the lidar measurements; in the middle stratosphere we can compare all measurements; and in the upper stratosphere we compare FTIR, Umkehr, MWR and lidar observations. Because of the lower average DOFS of the Umkehr measurements, the DOFS of the defined partial layers do not always reach one: we get on average 0.5, 0.6, 0.7, and 1.1 from troposphere up to the upper stratosphere. For MWR, we can calculate the DOFS for the partial layers from the averaging kernels as well to get average values of 1.1 in the middle stratosphere and 2.2 in the upper stratosphere.

Table 2 summarizes this division of partial columns and provides information on the DOFS and the median instrument uncertainties between 2001-2022. These uncertainties are divided into systematic uncertainties and random uncertainties (which includes the random smoothing error), which capture the accuracy and the precision of the measurement respectively. For the remote sounding measurements, the uncertainties within the partial column reported in Table 2 are obtained from the uncertainties contained in the error covariance matrix $\mathbf{S_x}$ of the measurement by

$$U_{\mathrm{PC}} = \mathbf{h^T S_x h}, \tag{3}$$

where $\mathbf{h}$ transforms the volume mixing ratio profiles to the appropriate partial columns, having values of zero at altitudes outside the boundaries of partial column. If a full error covariance matrix is not available, such as with the Umkehr observations, we use the available error estimates which we know in function of altitude. These values are then used to calculate the errors for partial columns by taking a weighted average, similar as described above.

### 3.2   Re-gridding, smoothing, and prior substitution

In order to perform a comparison between two measurements we need to take into account the different vertical resolution of the instruments. The passive, remote sounding instruments (FTIR, Umkehr, MWR) have different averaging kernels as well as a priori, which are both important to relate the retrieved state to the true state, as is seen in Equation (1). As already mentioned, we always perform the comparisons with respect to FTIR because these measurements provide total columns as well as partial columns from the surface up to 50km. Before integrating the partial columns to be compared, we transform the vertical profiles according to the vertical resolution following Rodgers and Connor (2003). We can distinguish two cases depending on the resolution of the compared instrument.

### First case: the compared instrument has a high vertical resolution: Sonde and lidar

In the FTIR retrievals the altitude grid is discretized into a lot fewer layers (47 for Lauder) than the high vertical resolution Sonde and lidar measurements. To proceed, we

**Table 2.** Total and partial column information showing the DOFS and mean systematic uncertainties ($U_s$), random uncertainties ($U_r$), random smoothing error ($U_{smooth}$), and total random error ($U_{r,tot} = \sqrt{U_r^2 + U_{smooth}^2}$) of all ground-based measurements. The Umkehr uncertainties are an estimate based on similar measurements in Boulder, USA.

| | Troposphere 0.5-11 km | | | | | Lower stratosphere 14-22 km | | | | |
| | DOFS | $U_s$ [%] | $U_r$ [%] | $U_{smooth}$ [%] | $U_{r,tot}$ [%] | DOFS | $U_s$ [%] | $U_r$ [%] | $U_{smooth}$ [%] | $U_{r,tot}$ [%] |
|---|---|---|---|---|---|---|---|---|---|---|
| FTIR | 1.0 | 9.1 | 0.7 | 6.9 | 6.9 | 1.0 | 7.2 | 1.1 | 3.5 | 3.7 |
| Umkehr | 0.5 | 7.5 | 1.6 | 2.7 | 3.1 | 0.6 | 4.0 | 2.0 | 1.1 | 2.3 |
| MWR | – | – | – | – | – | – | – | – | – | – |
| Lidar | – | – | – | – | – | – | $\sim$3 | 2.4 | – | 2.4 |
| Sonde | – | 0 | 7.3 | – | 7.3 | – | 0 | 4.3 | – | 4.3 |

| | Middle Stratosphere 22-29 km | | | | | Upper stratosphere 29-42 km | | | | |
| | DOFS | $U_s$ [%] | $U_r$ [%] | $U_{smooth}$ [%] | $U_{r,tot}$ [%] | DOFS | $U_s$ [%] | $U_r$ [%] | $U_{smooth}$ [%] | $U_{r,tot}$ [%] |
|---|---|---|---|---|---|---|---|---|---|---|
| FTIR | 0.9 | 3.5 | 2.6 | 2.7 | 3.7 | 1.0 | 9.6 | 2.5 | 2.2 | 3.3 |
| Umkehr | 0.7 | 3.0 | 0.4 | 0.7 | 0.8 | 1.1 | 5.0 | 0.6 | 1.2 | 1.3 |
| MWR | 1.1 | 3.6 | 2.0 | 2.5 | 3.2 | 2.2 | 2.9 | 2.2 | 1.2 | 2.5 |
| Lidar | – | $\sim$3 | 1.8 | – | 1.8 | – | $\sim$3 | 4.5 | – | 4.5 |
| Sonde | – | 0 | 4.2 | – | 4.2 | – | – | – | – | – |

| | Total column | | | | |
| | DOFS | $U_s$ [%] | $U_r$ [%] | $U_{smooth}$ [%] | $U_{r,tot}$ [%] |
|---|---|---|---|---|---|
| FTIR | 4.3 | 3.2 | 1.2 | 0.12 | 1.2 |
| Umkehr | 3.5 | 2.0 | 1.1 | 0.5 | 1.2 |
| Dobson | - | 5.0 | 1.0 | - | 1.0 |
| UV2 | - | 5.0 | 1.0 | - | 1.0 |

will first re-grid the measurement of higher resolution to fit the vertical grid of FTIR. For this re-gridding we follow the approach of Langerock et al. (2015) where the 'source' grid of the high-resolution observation is transformed to the FTIR grid. A transformation matrix is constructed which contains the fractions of how the FTIR grid is covered by this source grid, where the interpolation is constructed such that mass is conserved. In Figure 3 (a), an example is shown on how the ozone profile changes due to the re-gridding onto a lower resolution grid.

As already explained, the 47 FTIR layers however do not all provide accurate information on the ozone profile because of the limited DOFS of the instrument. In order to simulate this profile retrieval (as would be hypothetically observed by an FTIR measurement) for the higher resolution measurement, the latter is smoothed using the FTIR averaging kernel as described in Rodgers and Connor (2003). By setting the FTIR measurement as 'measurement 1' and the higher resolution measurement as '2', we can smooth the (by now re-gridded) profile $\mathbf{x}_2$ through

$$\mathbf{x}_2^{smooth} = \mathbf{x}_1^{apr} + \mathbf{A}_1 \left( \mathbf{x}_2 - \mathbf{x}_1^{apr} \right), \tag{4}$$

where $\mathbf{x}_2^{smooth}$ represents the smoothed profile of the higher resolution instrument with the FTIR averaging kernel $\mathbf{A}_1$ using the a priori profile of FTIR $\mathbf{x}_1^{apr}$. This smoothing step

is represented by the blue line in Figure 3 (a) to see the final ozonesonde measurement that is compared to FTIR. The final step is to extract the partial ozone columns by integrating the profiles using the defined partial layers above.

**Second case: the compared instrument has also a low vertical resolution: Umkehr and MWR**

For the remote sounding measurements (MWR and Umkehr) an additional step needs to be considered. These retrievals are typically made with different a priori, which we need to account for in the intercomparison. In the case of the MWR observations, before we smooth the MWR profile using the FTIR averaging kernel, we will transform it, as described in Rodgers and Connor (2003), by substituting the FTIR prior $\mathbf{x}_1^{apr}$ according to

$$\mathbf{x}_2^{sub} = \mathbf{x}_2 + (\mathbf{A}_2 - \mathbf{I}) \cdot (\mathbf{x}_2^{apr} - \mathbf{x}_1^{apr}), \tag{5}$$

with $\mathbf{I}$ a unit matrix and $\mathbf{x}_2^{apr}$ the MWR prior. This way the MWR retrieval is adjusted for the different a priori and the profile $\mathbf{x}_2^{sub}$ is subsequently used in Equation (4), instead of $\mathbf{x}_2$, for the intercomparison after smoothing with the FTIR averaging kernel.

In the case of Umkehr, the same transformation steps apply, however the DOFS are lower than for FTIR so the roles

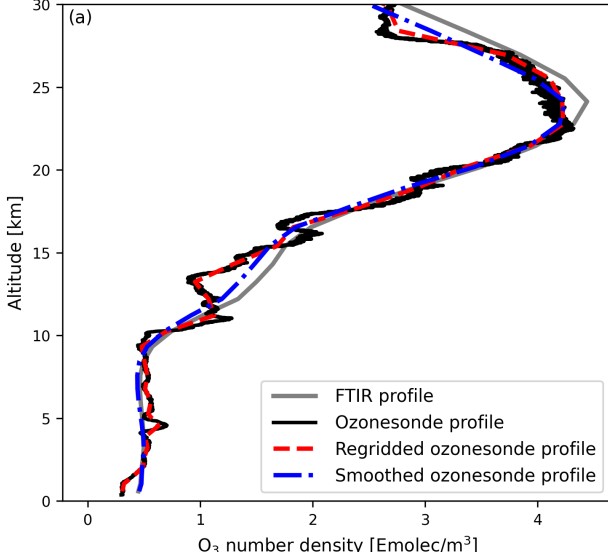

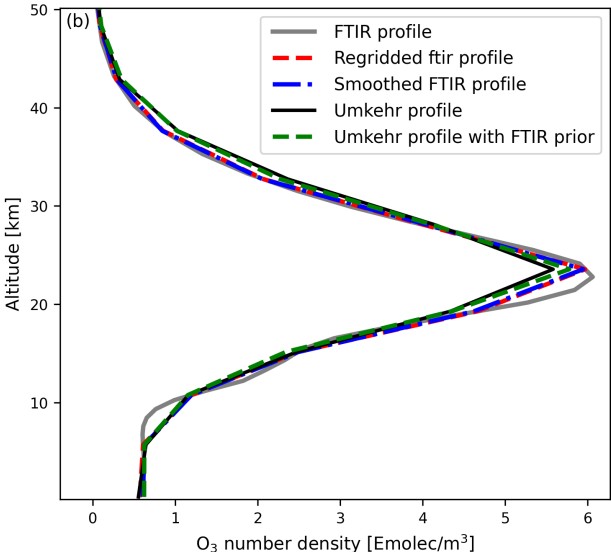

**Figure 3.** Various stages within the validation method, showing the re-gridding and smoothing of the ozone profile. Figure (a) shows one ozonesonde measurement in comparison with a near-simultaneous FTIR measurement. Figure (b) shows the profiles of Umkehr in comparison with FTIR, where additionally a step of the prior substitution is included.

are reversed. This means that the FTIR profiles are first re-gridded to the Umkehr grid (which has 16 vertical layers). Secondly, the Umkehr prior is substituted in the FTIR a priori by Equation (5) where now $\mathbf{x}_2^{\mathrm{apr}}$ is the Umkehr prior. Lastly, before calculating partial columns from the profile, the FTIR profile is smoothed using the Umkehr averaging kernel according to Equation (4), where again here measurements 1 and 2 refer to Umkehr and FTIR respectively. Figure 3 (b) shows the effect each of the transformation steps has on the FTIR and Umkehr profiles. Here we see that the biggest change comes from substitution of the FTIR prior in the Umkehr retrieval, which shifts the Umkehr profile closer to the FTIR profile at the maximum of the ozone number density

## 3.3 Time coincidence

The observations are not taken simultaneously, so a choice has to be made constructing pairs of observations. The construction is done by considering each separate measurement of Umkehr, MWR, lidar, or ozonesonde, and finding the FTIR measurement which lies within a time window around the observing time of the other instrument, where the window itself depends on the instrument we compare to. The choice of the time windows is elaborated on in Appendix B. By choosing observation pairs as opposed to the full data sets, we automatically select the same time sampling of both instruments when calculating the bias and long-term drift. This leads to a time window of 6 hours for Umkehr, ozonesondes, Dobson and UV2, 12 hours for lidar, and 3 hours for MWR.

If more than one FTIR measurement falls within the time window of another observation, all these FTIR measurements are averaged. Because we do this, the random uncertainty associated with the measurements will be reduced. If there are $N$ number of measurements within the comparison window, then the random error is reduced by $\sqrt{N}$ for this particular comparison. If both measurement techniques have multiple measurements in the same day that fall within the time window (for example, one or more FTIR measurements are taken within three hours from two different MWR measurements), then the FTIR measurements that fall in this overlap are used for comparison in both observation pairs.

## 4 Results

### 4.1 Bias and dispersion analysis

Now that we have defined the validation setup to perform the intercomparison, we can analyze the time series of the various ground-based measurements compared to FTIR. The first metric for our validation between the ground-based instruments is the bias. The bias $M$ that we will report here is the median value (where we choose the median over the mean, because of its robust nature with respect to outliers) of

the full time series of the relative differences

$$\Delta_{\mathrm{rel}} = \frac{PC_{\mathrm{X}} - PC_{\mathrm{FTIR}}}{PC_{\mathrm{FTIR}}} \cdot 100\% \quad (6)$$

$$M = \mathrm{med}\left(\Delta_{\mathrm{rel}}\right), \quad (7)$$

where we consider the total or partial column ($PC$) of FTIR and a second measurement X. Before we use the relative differences to produce the results, we filter for outliers which fall beyond the $3\sigma$ deviation. Additionally, to analyze the precision within the intercomparison, we use the MAD (Median Absolute Deviation), which is the median of the absolute deviations from the overall median.

$$\mathrm{MAD}_s = 1.4826 \cdot \mathrm{med}\left(\mathrm{abs}\left(\Delta_{\mathrm{rel}} - M\right)\right). \quad (8)$$

Here is seen that we always use the scaled $\mathrm{MAD}_s$ with a constant factor 1.4826. This scaling factor makes the MAD representative as a deviation from the median, similarly as the standard deviation is to the average, in the case of a normal distribution (Rousseeuw and Croux, 1993). We are not dealing with perfect Gaussian distributions, but the factor still creates a reasonable value for the scatter. The $\mathrm{MAD}_s$ is thus similar to using the standard deviation but is more robust in the sense that it will be less affected by outliers. In order to put the bias and $\mathrm{MAD}_s$ in perspective, we will compare these to the combined systematic and random uncertainties of the two involved instruments respectively. These combined uncertainties are calculated by

$$\sigma_{\mathrm{comb}} = \sqrt{\sigma_{\mathrm{FTIR}}^2 + \sigma_{\mathrm{X}}^2} \quad (9)$$

where $\sigma$ can signify either the systematic or random uncertainty. However, if we are dealing with the comparison of two remote sounding instruments (Umkehr and MWR), the simple combination of Equation (9) is not sufficient, because we also need to account for the smoothing error from the retrieval. Because the averaging kernels are not unit matrices, the total combined errors will be correlated. From Rodgers and Connor (2003), we find the covariance of the difference $\mathbf{S}_{\mathrm{comb}}$ to be

$$\mathbf{S}_{\mathrm{comb}} = \left(\mathbf{A}_{\mathrm{FTIR}} - \mathbf{A}_{\mathrm{FTIR}}\mathbf{A}_{\mathrm{X}}\right)\mathbf{S}_{\mathbf{a}}\left(\mathbf{A}_{\mathrm{FTIR}} - \mathbf{A}_{\mathrm{FTIR}}\mathbf{A}_{\mathrm{X}}\right)^T$$
$$+ \mathbf{S}_{\mathrm{FTIR}} + \mathbf{A}_{\mathrm{FTIR}}\mathbf{S}_{\mathrm{X}}\mathbf{A}_{\mathrm{FTIR}}{}^T. \quad (10)$$

This equation includes the separate errors on the covariances $\mathbf{S}_{\mathrm{FTIR}}$ and $\mathbf{S}_{\mathrm{X}}$ as the last two terms and includes the different averaging kernels $\mathbf{A}_{\mathrm{FTIR}}$ and $\mathbf{A}_{\mathrm{X}}$ which account for the smoothing error of both retrievals on the a priori covariance matrix $\mathbf{S}_{\mathbf{a}}$.

For every pair of comparisons we give the numbers of the median and scaled $\mathrm{MAD}_s$ of the partial-column relative differences as well as the combined random and systematic uncertainties in Table 3. Also the Pearson correlation coefficient between the individual time series in time coincidence (see Section 3.3) is shown as a measure of the measurements'

agreement and their capacity to capture the ozone variability. We also give the Pearson correlation between the monthly anomalies of each time series in order to remove the strong seasonality which could drive the correlation. The anomalies are constructed by taking a relative difference of the ozone column with the mean ozone column of the same month over the full time series. The last column shows the amount of observation pairs that are used for the intercomparison, indicating the reduced sample size compared to the full time series of the observations. The results of this table will be discussed in the following sections where we consider the total column and each partial column defined above separately, reporting on these biases and dispersion.

### 4.1.1 Total column

The correlation between the FTIR and Dobson total column anomalies shows an excellent Pearson correlation coefficient of 0.98 for individual time series in coincidences, and 0.97 for the monthly anomalies. Furthermore, the bias and dispersion of -2.9% and 2.0%, respectively are small and well within the combined systematic and random uncertainties respectively (see Table 3). Because the Dobson spectrometer that gives us the TCO measurement is the same as used for the Umkehr data, we do not have an independent comparison, especially the Umkehr retrieval is performed such as its integrated column is the same as Dobson TCO within the uncertainty. However, we still do the comparison for completeness, obtaining indeed the same bias (-2.9%), but slightly smaller dispersion (1.7%). We see that is nicely in agreement with the smaller combined random uncertainty budget (2.3% for Dobson TCO, 1.7% Umkehr), and is therefore due to the different sampling of the Dobson and Umkehr measurements (see Figure 1, and number of pairs in Table 3). If we apply the Dobson TCO comparisons with the same sampling as the Umkehr, the scatter indeed decreases down to 1.8%. These variations provide an empirical proof that the random uncertainties budget of FTIR and Dobson total columns are very well estimated, and reach indeed a precision of 1.2% and 1.0%, respectively.

Considering the bias, a larger value of -4.5% was found between FTIR and Brewer observations in the past at Izaña (Schneider et al., 2008). The improvement to a smaller bias in the present study is largely thanks to the change in spectroscopic input parameters of the FTIR retrieval strategy (see Section A2), where we now use the HITRAN2020 database, in which the ozone infrared line intensities in the FTIR retrieval spectral region have proven to be more in agreement with the UV cross-section than previous HITRAN versions (Gordon et al., 2022) that were used in the IRWG community.

Although the improvement is clear, we still obtain a slight overestimation of the FTIR columns, which is confirmed by the bias obtained with the UV2 measurements (-1.8%), pointing to a residual slight bias between both infrared and UV

**Table 3.** Results of the validation for the partial columns which show relative differences $\frac{\mathrm{X-FTIR}}{\mathrm{FTIR}}$ of the measurements (X) with respect to FTIR. The median of the entire time series, or bias $M$, as well as the scaled MAD$_s$ are shown. For every partial column we also show the combined systematic and random uncertainties of the involved measurements in percent. We show the Pearson correlation coefficient $r_{\mathrm{indiv}}$ between the individual time series of FTIR and measurement X as well as the correlation $r_{\mathrm{anom}}$ between their monthly anomalies to remove the effect of seasonality. The last column shows the number of coincident pairs of the intercomparison to indicate the reduced sampling from the total time series.

| | Bias $M$ [%] | MAD$_s$ [%] | $\sigma_{\mathrm{sys}}^{\mathrm{comb}}$ [%] | $\sigma_{\mathrm{rand}}^{\mathrm{comb}}$ [%] | $r_{\mathrm{indiv}}$ | $r_{\mathrm{anom}}$ | Number of pairs |
|---|---|---|---|---|---|---|---|
| Sonde | | | | | | | |
| 0.5-11 km | -1.9 | 6.8 | 9.1 | 7.3 | 0.91 | 0.78 | 1059 |
| 14-22 km | -6.8 | 4.6 | 7.2 | 4.4 | 0.97 | 0.92 | 1010 |
| 22-29 km | -6.4 | 4.0 | 3.5 | 4.9 | 0.86 | 0.76 | 1016 |
| Umkehr | | | | | | | |
| 0.5-11 km | 10.7 | 17.9 | 11.7 | 7.6 | 0.78 | 0.48 | 3694 |
| 14-22 km | -3.2 | 5.2 | 8.1 | 4.3 | 0.93 | 0.86 | 3622 |
| 22-29 km | -6.6 | 4.7 | 4.5 | 3.8 | 0.83 | 0.64 | 3685 |
| 29-42 km | -1.9 | 5.0 | 9.9 | 3.6 | 0.57 | 0.61 | 3683 |
| TCO | -2.9 | 1.7 | 3.2 | 1.7 | 0.98 | 0.97 | 4447 |
| Lidar | | | | | | | |
| 14-22 km | -1.2 | 8.3 | 7.8 | 2.6 | 0.94 | 0.79 | 1316 |
| 22-29 km | -5.2 | 4.0 | 4.6 | 3.2 | 0.78 | 0.83 | 1359 |
| 29-42 km | -1.7 | 4.8 | 10.1 | 5.1 | 0.87 | 0.75 | 1260 |
| MWR | | | | | | | |
| 22-29 km | -5.7 | 5.3 | 6.2 | 4.9 | 0.78 | 0.73 | 2679 |
| 29-42 km | -1.4 | 5.9 | 10.5 | 5.2 | 0.70 | 0.74 | 2661 |
| Dobson | | | | | | | |
| TCO | -2.9 | 2.0 | 3.2 | 2.3 | 0.98 | 0.97 | 5223 |
| UV2 | | | | | | | |
| TCO | -1.8 | 2.9 | 3.4 | 2.4 | 0.94 | 0.93 | 2880 |

spectroscopies. The agreement between FTIR and UV2 is also very good, with a correlation of the monthly anomalies of 0.93, and a dispersion of 2.9%. However, this dispersion is higher than the combined random uncertainty budget. To explain this, we compare Dobson and UV2 measurements directly to each other which are taken within a time window of 12 hours. The dispersion between the two instruments is found to be 1.9%. This is higher than the 1% random uncertainty of the UV2 instrument which indicates that the random error budget of UV2 is underestimated.

### 4.1.2 Troposphere

In the tropospheric column between the ground ($\sim$0.5 km) and 11 km, we compare the FTIR observations with both Umkehr and sonde data.

The FTIR and sonde tropospheric columns are in very good agreement as shown by the high Pearson correlation coefficients of 0.91 and 0.78, for individual coincidences and monthly anomalies respectively, and by the small bias of -1.9% and the dispersion of 6.8%, well within the combined uncertainties. A negative bias is expected due to remaining systematic bias in the infrared spectroscopy.

The situation is less favorable with the Umkehr comparisons. While the Pearson correlation with individual time series is good (0.78), it decreases down to 0.48 for the monthly anomalies. This is also seen in the much larger dispersion (17.9%), which is larger than the random uncertainty budget. The observed bias is also large and positive (10.7%), although still within the systematic uncertainty budget. The DOFS of the Umkehr tropospheric columns (0.5-11km) is only 0.5, which can explain the weaker agreement there.

We give in Appendix C the effect of the a priori substitution and smoothing procedure (Section 3.2) on the profile comparisons. It is seen in Figure C1, that the Umkehr profile in the troposphere is the only place where the procedure is worsening the comparisons compared to direct ones. Therefore, we give here also the numbers for the FTIR and Umkehr tropospheric columns direct comparisons without smoothing: the bias is better with only 1.8%. The fact that this bias between FTIR and direct Umkehr tropospheric ozone is small is interesting for use in HEGIFTOM studies at Lauder. However the scatter stays at similar levels (18.1%) as well as the correlation: $r_{\mathrm{indiv}}$ = 0.82, $r_{\mathrm{anom}}$ = 0.34. Because the FTIR and ozonesonde dispersion agree well within the combined random uncertainty, this large scatter between FTIR and Umkehr points to an underestimation of the Umkehr ran-

dom uncertainty budget, which presently only accounts for the measurement noise.

### 4.1.3 Lower stratosphere

In the lower stratosphere (14-22 km) we can again compare FTIR to Umkehr and sonde data and additionally the lidar measurements which start above the tropopause. There is a strong correlation for these partial columns, between 0.93 and 0.97 for the individual time series, and still between 0.79 and 0.92 for the monthly anomalies. For all comparisons we find a negative bias from -1.2% for lidar, -3.2% for Umkehr, and -6.8% for sonde. Similarly as with the results for the troposphere, all these values fall within the range of the combined systematic uncertainties for the respective instruments in the intercomparisons, and the systematic bias of FTIR is confirmed. For the dispersion of the FTIR and sonde comparison, we find a value of 4.6%, which is very comparable to the combined random uncertainty. For Umkehr we get a dispersion of 5.2%, which is only slightly larger than the random instrument uncertainties in this partial column (4.3%). For lidar, the dispersion (8.3%) is significantly larger than the uncertainty (2.6 %). We should keep in mind that the time coincidence is set to 12h between the FTIR and lidar that are measuring during day and night, respectively. Therefore, a collocation mismatch could play a larger role in this comparison.

### 4.1.4 Middle stratosphere

In the middle stratosphere (22-29 km) we have all five ground-based measurements available for intercomparison. We find a good correlation between the measured partial columns, between 0.78 and 0.86 for individual time series, but decreasing to 0.64 and 0.83 for the monthly anomalies, the worse correlation being with Umkehr. For all measurements we find a negative bias of around -5 to -7% with respect to FTIR. The bias found with MWR falls within the combined systematic uncertainties in this column. However, with respect to the ozonesonde, Umkehr, and lidar data, the bias is greater than the combined measurement uncertainties. A similar bias of -7% was found in García et al. (2012) between FTIR and sonde at Izaña in this partial column, meaning that the improvement of the infrared spectroscopy (Gordon et al., 2022) since this study, which was using a previous version of HITRAN, is not sufficient to reduce the FTIR systematic bias in the middle stratosphere. Apart from the largest contribution of FTIR systematic bias which is still the spectroscopy (3.1%), the uncertainty due to temperature reaches its highest value in this partial column, aside from the region above 40 km. An underestimation of the temperature uncertainty used in our theoretical estimation of error budget at these altitudes would lead to an underestimation of the FTIR systematic bias.

The dispersions are however small in the middle stratosphere

with values between 4.0 and 5.3%, within or only slightly larger than the combined random uncertainties.

### 4.1.5 Upper stratosphere

In the upper stratosphere (29-42 km) we can no longer compare to the sonde data, but we still have the other measurements available for intercomparison. The correlation coefficient for the partial columns are not as good in this partial column, with values between 0.61 and 0.75 for the monthly anomalies, but we should keep in mind that ozone shows much less variability at this altitude range. None of the measurements show a large bias with respect to FTIR where the median value is between -1 and -2% with a dispersion of 4-6%. So considering respectively the systematic and random instrument uncertainties, the four measurements are in agreement. Again, even if small, the bias seems to be in FTIR data only, confirming what is observed with total and other partial columns measurements, pointing to a remaining small bias in the infrared spectroscopic parameter in HITRAN2020.

### 4.2 Drift analysis

In our study, we also want to obtain the difference in total and partial column trends between the measurements (drift). In the analysis of these relative differences between two measurements within one observation pair, we simply apply a linear fit to the time series. Deseasonalizing of the time series is not necessary, because we are dealing with differences of ozone measurements at the same location, so the ozone variability is cancelled out. The slope of the linear trend in the relative differences quantifies the drift. We fit the relative differences between monthly means of data in coincidence $Y(t)$

$$Y(t) = A_0 + A_1 t \tag{11}$$

with $A_0$ the intercept, $A_1$ gives the drift, and time $t$ is given in fractional years. The monthly means are first calculated from the two measurements separately after which we take the relative difference, as in Equation (6). We have tested to include a seasonal cycle in the regression analysis, in case a seasonal cycle in the differences would appear, but it turns out to be barely significant and without impacting the obtained drift themselves. The 2-$\sigma$ trend error obtained from the fit is corrected with the auto correlation of the residuals, according to Santer et al. (2008). This results in a higher uncertainty if any correlation remains in the residuals of the fit. This trend uncertainty has the form

$$U_{\text{drift}} = 2\sigma_{A_1} \sqrt{\frac{N-2}{N_{\text{eff}}-2}}, \tag{12}$$

with $\sigma_{A_1}$ the standard deviation on the fit parameter $A_1$, $N$ the degrees of freedom (not to be confused with DOFS from a retrieval) being the length of the monthly-means time series, $N_{\text{eff}} = N\frac{1-R}{1+R}$ in which $R$ is the correlation coefficient

between the residuals of two consecutive time steps. This error will be used to express our confidence in the drift values obtained in the results. Here 'significant' actually refers to a high certainty of the drift value on a 95% confidence interval. Similarly, 'non-significant' refers to only a medium to very low confidence in the obtained result. This terminology will be used from now onwards in the paper.

We summarize the obtained drift and uncertainty for each measurement with respect to FTIR in Table 4. Additionally, Figures 4 and 5 shows the time series of the relative differences within the intercomparison together with the trend analysis. Figure 6 visualizes the obtained drifts in absolute units for the total column as well as all partial columns. These values are obtained by performing the same trend fitting procedure on the absolute differences of the monthly means in Dobson Units (DU).

### 4.2.1 Total column

When we apply the fit from Equation (11) to the relative differences of FTIR and Umkehr we obtain a very similar drift between FTIR and Dobson (0.4 %/decade) and between FTIR and UV2 (0.4 %/decade), although the periods are different. These two drifts are however not significant considering their uncertainties ($U_{\mathrm{drift}} = 0.4$ %/decade and 1.8 %/decade, respectively). If we calculate the drift with Dobson for the same time period as for the UV2, we obtain $0.3\pm1.1$ %/decade. We also give in Table 4 the drift obtained between FTIR and Umkehr total columns ($0.6 \pm 0.4$%/decade), however this is a redundant result as already discussed, the Umkehr columns being constrained by the Dobson ones. Furthermore, as seen in Figure 1 and Table 3, the sampling is smaller with Umkehr, which however does not impact the uncertainty on the drift.

The small values of the drifts (and their insignificance) prove the very good stability of the three total columns ozone measurements at Lauder.

### 4.2.2 Troposphere

We find non-significant drifts of -0.4$\pm$6.1 %/decade between Umkehr and FTIR and -0.1$\pm$2.3 %/decade between ozonesonde and FTIR, which are shown in Figure 4.

We should note that the drift uncertainty is quite large (6.1 %/decade) between the two low-resolution instruments, which is not surprising given the high dispersion value found between both tropospheric measurements in previous section (18%, see Table 3). This uncertainty is lower (4.6 %/decade) if we perform the drift analysis on direct partial column comparisons (without performing the smoothing), as detailed in Appendix C.

However, both of these drifts are small and non-significant within the uncertainty, so we can say with strong confidence that there is no drift between the FTIR, Umkehr and ozonesonde tropospheric ozone data sets in the past two decades. These results prove that at Lauder, we have consistent long-term tropospheric ozone measurements from three independent ground-based measurements, suitable for trend studies as planned for TOAR-II.

### 4.2.3 Lower stratosphere

After performing the intercomparison of the four measurements in this partial column the results show non-significant drifts of -1.0$\pm$1.4 %/decade for FTIR with the sonde data and 0.4$\pm$2.7 %/decade for FTIR with the lidar data. We do, however, find a positive drift of Umkehr with respect to FTIR of 2.6$\pm$1.1 %/decade. In Godin-Beekmann et al. (2022), the Umkher trend at Lauder was indeed the only one showing a positive trend at this altitude, not only compared to other ground-based measurements at Lauder, but also to a set of satellite overpasses. In Section 5 we will elaborate more on how these drifts align with the observed stratospheric trends in Godin-Beekmann et al. (2022).

### 4.2.4 Middle stratosphere

The resulting drifts between FTIR and the other measurements are 1.5$\pm$1.7 %/decade with Umkehr and 0.1$\pm$1.4 %/decade with ozonesonde, both non-significant, and two positive drifts of 2.0$\pm$0.8 %/decade with lidar and 3.1$\pm$2.1 %/decade with MWR. The study of Bernet et al. (2020) also considers both lidar and MWR as two ground-based measurements in the study of stratospheric ozone trends at Lauder. These trends agree to within 2 to 3 % with each other, which corresponds to similar numbers in our intercomparison. Unfortunately, for lidar and MWR, we do not have full time series available to have the same period of analysis as for the Umkehr and ozonesonde comparisons As mentioned above, the lidar measurements are missing 3 years of data from 2012-2015. This incomplete sampling may influence the real lidar trend and potentially the drift with FTIR. Additionally, the MWR data only extends to 2016 at Lauder.

### 4.2.5 Upper stratosphere

We find negative drifts for Umkehr -3.2$\pm$1.2 %/decade with FTIR and for lidar -4.0$\pm$1.2 %/decade with FTIR, and we find a non-significant drift of MWR -1.7$\pm$2.1 %/decade with FTIR. As mentioned for the drift in the middle stratosphere, the incomplete time series of the lidar data could lie at the origin of the magnitude of this drift. While in this partial column, the conclusions concerning the drift for lidar and MWR with FTIR differ from one another in magnitude, the values are still in agreement with Bernet et al. (2020), where both measurements have small positive trends in the upper stratosphere around 2 to 3 %/decade with MWR peaking at slightly larger trends in the upper stratosphere, which agrees with the smaller drift.

**Table 4.** Drifts with respect to FTIR for each of the measurements in %/decade for every partial column where they have data available. The drifts that are significantly different from zero (considering $U_{\mathrm{drift}}$) are represented in boldface.

| Drifts [%/decade] | Troposphere 0.5-11 km | Lower stratosphere 14-22 km | Middle stratosphere 22-29 km | Upper stratosphere 29-42 km | Total column |
|---|---|---|---|---|---|
| Umkehr 2001-2022 | -0.4 ± 6.1 | **2.6 ± 1.1** | 1.5 ± 1.7 | **-3.2 ± 1.2** | **0.6 ± 0.4** |
| Sonde 2001-2022 | -0.1 ± 2.3 | -1.0 ± 1.4 | 0.1 ± 1.4 | – | – |
| Lidar 2001-2011,2015-2022 | – | 0.4 ± 2.7 | **2.0 ± 0.8** | **-4.0 ± 1.2** | – |
| MWR 2001-2016 | – | – | **3.1 ± 2.1** | -1.7 ± 2.1 | – |
| Dobson 2001-2022 | – | – | – | – | 0.4 ± 0.4 |
| UV2 2012-2022 | – | – | – | – | 0.4 ± 1.8 |

### 4.2.6 Drift discussion

The significant drifts obtained in the stratospheric columns will unfortunately lead to different long-term stratospheric ozone trends from the different instruments at Lauder. While when comparing only two different techniques, we can not distinguish which of the two (or if the two) might have stability issues, our study using multiple instruments seems to point to the Umkehr time series for the lower stratosphere, and more clearly to the FTIR measurements at least as one of the drift responsible in the middle and upper stratosphere.

We therefore tried to explore possible reasons for the drift in FTIR stratospheric time series, tracking instrument and data processing changes. Related to these instrument changes, Appendix D gives a history of calibration for both the FTIR and Dobson instruments. First, it should be noted, that the new FTIR retrieval strategy used in this study (Section 2.1 and Appendix A1) already improved significantly the drifts that would be obtained if the past IRWG version would be used, especially in the troposphere and upper stratosphere. This improvement is explained in more details in Appendix A2, and is mainly due to a change in regularization. Because drifts are still present with the improved FTIR data sets, we had first a look on the stability of the signal to noise ratio of the retrievals but they are constant over the full time period. Another influence on the drift could be a discontinuous step in the times series caused by instrumental factors, which we discuss in detail in Appendix E. Here we find that a step in May 2018 is present in several time series of differences between FTIR and other measurements, and in the FTIR DOFS time series. This step is identified in the FTIR log to a date with major instrument alignment. Such a step as found in FTIR shows the need for a detailed instrument log of all instruments to perform similar change-point analyses. While taking in to account the step in the FTIR data does reduce the drift slightly, it is still present and the reason for this drift and those in the other partial columns still requires some more in-depth analysis. Because the temperature is one of the main driver of FTIR systematic bias for the middle-upper stratospheric ozone, one should have a closer look in the future on a possible small drift of the NCEP temperature profiles used in the IRWG retrievals.

To better understand these drifts, one should also notice that while Umkehr (although non-significant here), lidar and MWR show all positive drifts with FTIR in the middle stratosphere, the drift is almost zero with the ozonesonde time series. We address here the TCO dropoff for these ozonesonde measurements found in certain sites, to see how the signal found at Lauder could affect the results. If we would assume a drift of 3%/decade in the Lauder ozonesonde time series, then correcting it would propagate to a increase in the drift with respect to FTIR to a value of 1%/decade in the lower stratosphere and 2%/decade in the middle stratosphere, bringing these values closer to the drift found with lidar. While it is decided in the present study to use the ozonesonde data sets from HEGIFTOM that follow the WMO/GAW 2021 homogenization procedures (See Section 2.5), we have performed as a test the drift study on the ozonesonde data set on which the Time Response Correction + Calibration (TRCC) method, as described in Smit et al. (2024) has been applied. The bias and dispersion with FTIR are worsening with this newly processed ozonesonde data set. In the middle stratosphere, where the impact of the new correction is largest, we see a bias of -9.3% and a scatter of 4.3%. However, it should be mentioned that the drift with FTIR is significantly positive (1.3±1.1 %/decade). This effect is very small in the case of Lauder (1.2 %/decade), but does seem to go in the good direction towards the other ozone stratospheric trend measurements at Lauder where we see a similar positive drift of the measurements with respect to FTIR. To con-

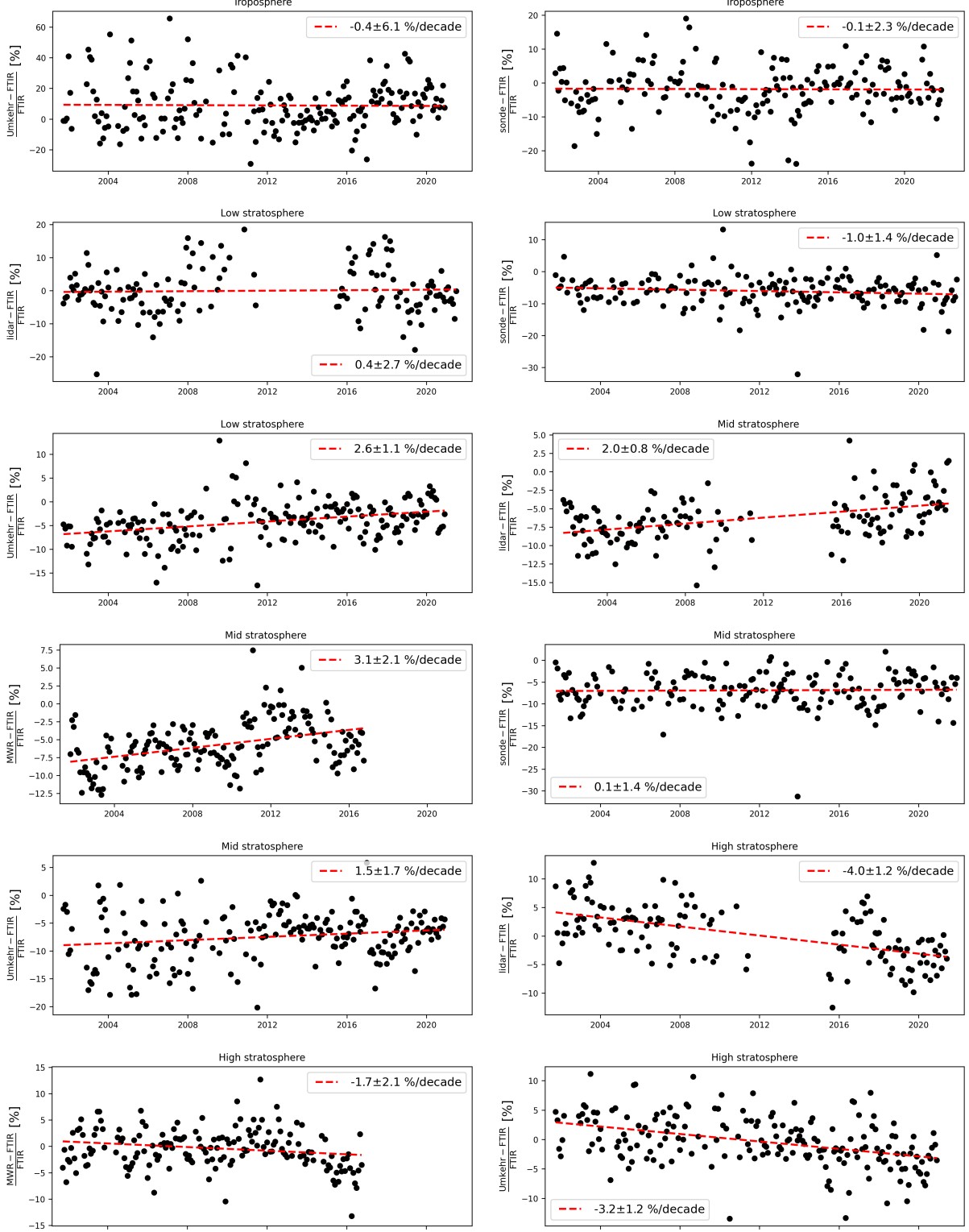

**Figure 4.** Relative differences between monthly means of sonde, Umkehr, MWR, and lidar with monthly mean of FTIR in the tropospheric and three stratospheric partial columns. Additionally, the linear trend fitted to the data is shown with the slope of this linear trend (the drift) and the trend error.

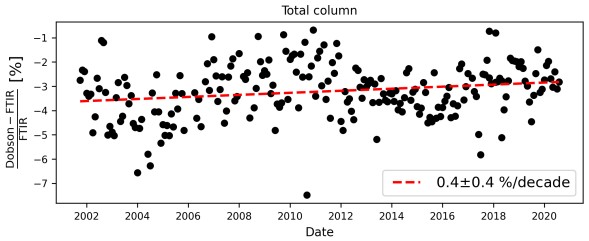
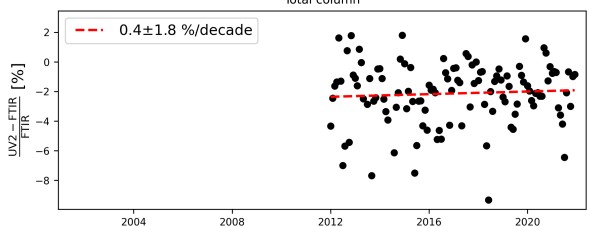

**Figure 5.** Same as Figure 4 but for total column comparison for Dobson and UV2.

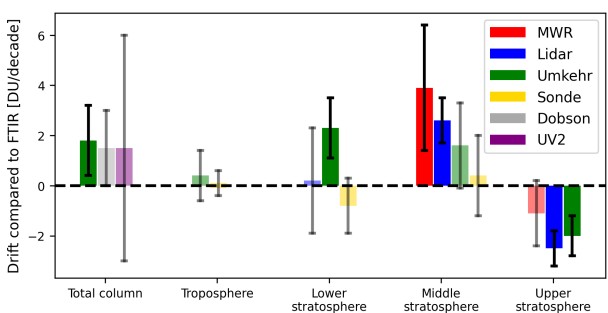

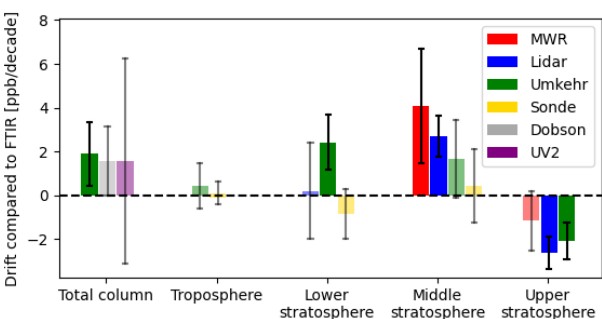

**Figure 6.** For the total column and each of the partial columns, the drift in DU/decade with respect to FTIR is shown for the Umkehr, ozonesonde, lidar, Dobson, and MWR data in the top figure. Additionally, the error margins from the trend analysis are shown. Non-significant trends are shown in a pale color for distinction. In the bottom figure, we show the same drifts but in molar fraction (ozone over dry air) per decade with units of parts per billion.

firm the changing trend when applying the TRCC method, we perform a similar drift analysis of the ozonesonde data sets to the lidar measurements. In the middle stratosphere we see that the drift (when using $\frac{\text{lidar} - \text{sonde}}{\text{sonde}}$) changes from 2.0±1.3 %/decade for the original ozonesonde data set to 0.7±1.6 %/decade for the newly processed data set. This seems to be consistent with the earlier results comparing sonde to FTIR, because the significant drift between lidar and sonde that is present with the original sonde data is not there for the newly processed sonde data, putting the trends of the ozonesondes more in line with that of lidar.
However, the new ozonesonde data processing methodology

used here is still in experimental phase, based on simulation chamber data, and should be assessed globally before implemented widely in the ozonesonde network. The intercomparisons done here subscribes the high potential of this new method.

## 5 Comparison to the LOTUS trend-analysis study (Godin-Beekmann et al., 2022)

One of the main aims of this work is to study if the discrepancy on the stratospheric ozone trends obtained at Lauder within LOTUS22 Godin-Beekmann et al. (2022) can be explained by intercomparing the measurements directly and checking their drifts.
One thing to note first is that the FTIR data used in their study uses the FTIR V08 (Vigouroux et al., 2008) retrieval strategy, so in order to make a sensible comparison, we should also give the FTIR V08 drift results. Furthermore, the FTIR trends are given for partial columns (both here and in Godin-Beekmann et al. (2022)) and not for the profile such as is the case for the other measurements in Godin-Beekmann et al. (2022). Additionally, the FTIR partial column altitude limits were also slightly different in this study than ours. In addition, there is a small difference between the time period used to derive the trends, which is 2000-2020 for Godin-Beekmann et al. (2022). The effect of these small differences on FTIR trends can be seen in Table 5 by looking at FTIR V08 and GB22 FTIR V08. This table shows the trends calculated for our partial columns using a similar LOTUS regression model as in Godin-Beekmann et al. (2022). Figure 7 visualizes this same information in a comparable way to how the results are presented in Figure 4a of (Godin-Beekmann et al., 2022), except that we show them for the same partial columns for all instruments, and that we have added the MWR trends, even if it should be kept in mind that they are for the 2000-2016 period, while the other measurements cover 2000-2021.
The comparison allows us to, on the one hand, see the differences between using trends in the profile or using trends in the partial columns; and on the other hand we can find the effect of sampling on the trends. The latter matters because Godin-Beekmann et al. (2022) use all individual mea-

**Table 5.** Trends for each of the observation measurements in %/decade for every partial column where they have data available. For FTIR, we show the trends derived with both the V08 (Vigouroux et al., 2008) and new retrieval strategies and also selecting only the 2000-2016 time period to calculate the trend. The FTIR V08 trends from Godin-Beekmann et al. (2022) (GB22) are additionally shown with their relevant altitude ranges.

| Trends [%/decade] | Troposphere 0.5-11 km | Lower stratosphere 14-22 km | Middle stratosphere 22-29 km | Upper stratosphere 29-42 km |
|---|---|---|---|---|
| FTIR V08 | $3.9 \pm 1.8$ | $-3.6 \pm 2.1$ | $-3.2 \pm 1.4$ | $4.2 \pm 1.1$ |
| FTIR V08 (2000-2016) | | | $-1.9 \pm 2.0$ | $4.1 \pm 1.6$ |
| GB22 FTIR V08 | – | $-4.5 \pm 2.7$ | $-1.7 \pm 1.2$ | $5.0 \pm 1.1$ |
| | – | $(12 - 20$ km$)$ | $(20 - 29$ km$)$ | $(29 - 49$ km$)$ |
| New FTIR | $1.1 \pm 1.7$ | $-3.5 \pm 2.0$ | $-2.8 \pm 1.0$ | $3.1 \pm 1.1$ |
| Umkehr | $1.9 \pm 4.0$ | $1.7 \pm 1.7$ | $-0.4 \pm 1.6$ | $0.0 \pm 0.9$ |
| Sonde | $1.5 \pm 2.2$ | $-5.1 \pm 2.1$ | $-2.6 \pm 1.2$ | – |
| Lidar | – | $-3.0 \pm 2.7$ | $1.1 \pm 1.3$ | $1.8 \pm 2.0$ |
| MWR | – | – | $2.3 \pm 1.6$ | $3.2 \pm 1.5$ |

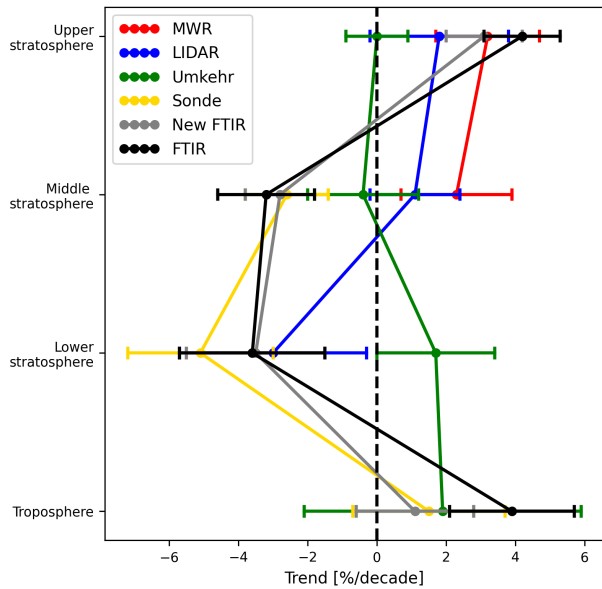

**Figure 7.** Figure showing the partial column trends of the different measurements at Lauder.

surements to calculate monthly means before deriving trends while in our drift calculation we make a selection of time coincidences in the intercomparison.

We find similarities in our current trends in Figure 7 and the ones from Godin-Beekmann et al. (2022): the Umkehr trend is still an outlier in the lower stratosphere, lidar still disagrees with sonde in the middle stratosphere while Umkehr, in the middle, agree with both. This means that the use of partial columns and the vertical resolution of the instruments (which here have been smoothed with the FTIR averaging kernels) does not impact the agreement/disagreement that

was observed in these altitude ranges in Godin-Beekmann et al. (2022). However, in the upper stratosphere, the use of partial columns makes the comparisons with lidar clearer, and the lidar trend is even now in agreement with FTIR which looked an outlier in Godin-Beekmann et al. (2022). This is due to partly to the fact that the lidar trend for this partial partial column (29-42 km) is 1.8%/decade while profile trends showed a mix of positive and negative trends within this large altitude range. The other reason for the FTIR to stop looking as an outlier is the decreased trend obtained with the new FTIR strategy (from 5% in Godin-Beekmann et al. (2022) to 3% in the current study), which is therefore a great improvement for LOTUS. We should also point out that the sonde trends are smaller in the present study than in Godin-Beekmann et al. (2022): -5.1 and – 2.6% in the lower and middle stratosphere respectively, while the values were between -7 and -3% in the latter study. However the ozonesonde data set is the same, pointing to a combined effect of partial column integration and vertical resolution and one additional year of data in the current study. Similarly as done before in Section 4.2.6 for the drifts, we can estimate the impact of a different ozonesonde data set if we would assume a correction for a 3% dropoff. This would result in an even smaller trend compared to LOTUS22, changing our trends from -5.1 to -2.1%/decade for the lower stratosphere and from -2.6 to 0.4%/decade in the middle stratosphere.

However, the agreement of our trends with LOTUS22 in the upper stratospheric trends of FTIR and lidar is not seen in our above drift studies since a significant drift of -4% was observed between them. We can clearly see here the influence of the temporal sampling. The gap in the lidar data and the poor collocation sampling can affect the drift significantly, and the uncertainty provided for that drift is probably underestimated and would require more sophisticated calculation techniques than applied here.

In the lower stratosphere we see a positive drift of Umkehr with FTIR (2.6±1.1 %/decade). Because the trends of FTIR and Umkehr are different and even of opposite sign, this gives confirmation of the drift we find in the lower stratosphere. For the ozonesonde and lidar, the trends are in good agreement, which also reflects in the insignificant drifts obtained above. While the drift values (Table 4) correspond well to the observed trends (Table 5) for sonde and lidar, we observe that the drift is smaller than expected by the trend differences (=5.2%) between Umkehr and FTIR. To check if the differences in Umkehr trend could then be due to the different temporal sampling or the vertical resolution, we give in Table C1 the drift between Umkehr and FTIR without the smoothing, and a similar drift is obtained. Part of the different trend in the Umkehr might then be due to the different temporal sampling. Similarly in the middle stratosphere, the differences observed in the trends between all instruments (except the sondes which are in agreement with FTIR) and FTIR is reflected only by half of the expected value in the observed drifts.

## 6 Conclusions

Long-term measurements of ozone are important to study the recovery of stratospheric ozone as well as the trends of total and tropospheric ozone. Because many sites use different measurements to monitor these ozone trends, there is a need to validate ozone measurements against each other. In this study we take advantage of the multitude of measurements available at the Lauder station (FTIR, Umkehr, ozonesonde, lidar, MWR, Dobson, and UV2) to perform an intercomparison between these measurements. The study, performed in the context of the LOTUS and TOAR-II initiatives, aims to show the biases and drifts, which require more attention to explain the different observed trend between ground-based measurements in Godin-Beekmann et al. (2022). Additionally, within the HEGIFTOM working group of TOAR-II there is a need for an intercomparison, evaluating all biases and drifts between tropospheric ozone measurements which is supplied in this study for the ozonesonde, FTIR and Umkehr measurements.

The method we use applies a comparison between observations (following Rodgers and Connor (2003) for the intercomparison of measurements with different vertical resolution) by manipulating the profiles through prior-substitution, re-gridding, smoothing and finally division into partial columns to perform the comparison. These steps are necessary because of the differences in profile retrieval and vertical resolution of the observations. Additionally, we take care to select pairs of comparisons within a specific time window where multiple FTIR measurements (if present) are averaged before performing the comparison.

For each of the instruments and partial columns we find a good correlation with FTIR (from 0.64 to 0.97 for the monthly anomalies). For the total column, we even find a correlation of 0.97. This shows that, even though a bias or drift might be present, the agreement of the long-term ozone measurements between FTIR and the other measurements is strong, capturing the same variability in all partial layers. Only between Umkehr and FTIR in the troposphere we find a moderate correlation of 0.48, probably due to the lower DOFS of Umkehr in this partial column.

The metrics we use to analyze the intercomparison are the robust median bias and (scaled) $MAD_s$ to track the accuracy and precision of the observations. These are compared respectively to the combined systematic and random instrument uncertainties. We find good agreement of FTIR with Dobson on the total column concerning the bias (-2.9%) and $MAD_s$ (2.0%) values, similar to Brewer-FTIR comparisons by Schneider et al. (2008). The same overestimation of FTIR is found with respect to UV2 pointing to a slight bias between infrared and UV spectroscopy, but still within the instrument uncertainties.

In the troposphere we find a low bias of -1.9% with the ozonesondes, but there is a larger value of bias (10.7%) and $MAD_s$ (17.9%) with Umkehr due to the low DOFS we have in this column for Umkehr. Additionally, this can point to an underestimation of the Umkehr random uncertainty budget. When we look at a direct comparison of FTIR to Umkehr without smoothing we actually find a low bias of 1.8%. So despite the low DOFS of Umkehr, this is a promising result for the use of both FTIR and Umkehr in the troposphere.

In the lower stratosphere we consistently find a negative bias between -1.2 and -6.8% for all instruments with respect to FTIR, but all fall within the range of the systematic uncertainties. This is not the case, however, for all values of $MAD_s$ (between 4.6 and 8.3 %) which is potentially due to an underestimation of the random instrument uncertainties. Additionally, there is the role of a collocation mismatch between FTIR and lidar, which are taken during the day and the night respectively.

In the middle stratosphere we again seem to find a negative bias between -5.2 and -6.6%, pointing towards too high values for FTIR in this partial column not accounted for in the uncertainty budget. This is possibly related to temperature profile or the treatment of the instrument line shape (ILS). When we look at the uncertainties in this partial column, we notice a higher error from the temperature profile compared to other partial columns.

In the upper stratosphere none of the measurements show a bias larger than -2% with respect to FTIR. The four measurements are found to be in agreement in the upper stratosphere when considering the systematic and random uncertainties.

Generally, even though often the bias falls within the systematic uncertainty, we consistently find a small negative bias of all instruments with respect to FTIR. This points to a remaining bias in the infrared spectroscopic parameter in HITRAN2020.

We calculated measurement drift by performing a linear fit to the relative differences. This results in a small, but non-significant drift in the total column of 0.4±0.4 %/decade between FTIR and Dobson and similarly 0.4±1.8 %/decade between FTIR and UV2 showing a good stability of all total column measurements at Lauder.

For tropospheric ozone we find no significant drifts between FTIR, Umkehr and ozonesondes proving that these are consistent long-term measurements at Lauder suitable for trend studies such as planned in TOAR-II.

We do find measurement drifts in the lower and upper stratospheres for FTIR with Umkehr of 2.6±1.1 %/decade and -3.2±0.9 %/decade, with lidar in the middle and upper stratosphere of 2.1±0.8 %/decade and -3.7±1.2 %/decade, and with MWR in the middle stratosphere of 3.1±1.7 %/decade. The drifts of the measurements in the same direction could point to stability issues for Umkehr in the lower stratosphere and for FTIR in the middle and upper stratosphere. In part, this drift of FTIR is due to a discontinuity in the time series due to an instrument alignment. We also suspect that the temperature profiles from NCEP, which affect the FTIR retrievals, have a potential drift propagating through the retrievals that needs further study.

In comparisons of FTIR and ozonesonde we find no significant drift. This means that there is strong agreement between ozonesonde and FTIR over all partial columns. We did, however, find that when we use newly processed ozonesonde data (Smit et al., 2024), a drift appears in the middle stratosphere between FTIR and sonde that is in line with the results of the other instruments in this partial column. Future trend studies with this ozonesonde data in the stratosphere should be carried out when this new methodology has been globally assessed in the ozonesonde network as well as studies to figure out the nature of the TCO dropoff in certain ozonesonde stations that could impact their trend analyses.

When comparing the stratospheric trends of each of the partial column for every measurement, we see that most of the trends are in agreement with those found in LOTUS22 (Godin-Beekmann et al., 2022), showing that the approach of partial columns in this study does not change the result much from considering the profile itself. Only in the upper stratosphere the trend changes rapidly with altitude, we find a small difference due to this effect, which in fact reduces the discrepancy between the lidar and FTIR V08 trends at this altitude. These trends are, in turn, mostly in agreement with the drift. The reason for the discrepancy is in part due to the fact that we account for different sampling of the data by constructing the comparison pairs. One example here is the missing 3-year gap in the lidar data at 2012-2015 or the shorter time series of MWR which stops in 2016. So while remaining drifts are still present, our study explains roughly half of the differences in observed trends in LOTUS22 (Godin-Beekmann et al., 2022) by the different sampling, vertical sensitivity or time periods and gaps. Additionally, the improved FTIR data in the current work has reduced the differences in the upper and tropospheric trends since LOTUS22.

The good agreement of the three measurements in the troposphere (concerning no significant bias or drift) show that these are reliable to use for trend studies within HEGIFTOM. Future studies can take advantage of this by merging the FTIR, Umkehr and ozonesonde measurements in order to provide more accurate trends thanks to the higher sampling (Chang et al., 2024). However, because no strong correlation is found between Umkehr and FTIR in the troposphere and the Umkehr DOFS or consistently low here, one has to be careful in the inclusion of the Umkehr tropospheric column into the merged product.

The bias and drift teach us that we can have confidence in the ground-based measurements for trend studies at Lauder. However, some attention has to be given firstly to the bias of FTIR in the stratosphere (especially the middle stratosphere) pointing to an underestimation of the infrared spectroscopic line intensities; secondly to Umkehr in the troposphere where we see a lower correlation and high dispersion when compared to FTIR; and lastly for ozonesondes in the middle stratosphere where the newly developed TRCC correction results in a changed drift from the currently used correction for ozonesonde data. On top of this great care has to be given to the effect of temporal sampling, gaps, and jumps in the ozone timeseries, where we for example found an influence on the trend from a discontinuity in the FTIR data related to a major alignment.

Lastly, in Appendix A we have found that employing the new FTIR strategy reduces bias with respect to the other measurements by 2 to 3%, which is mostly thanks to the change to HITRAN2020 spectroscopy. Furthermore, the new strategy reduces (at least at Lauder) drifts present in the FTIR V08 data thanks to new regularization resulting in an overall agreement of FTIR with ozonesondes.

*Data availability.* The ground-based data sets used in this article are collectively available in the following depository: https://doi.org/10.18758/as5rz1oh. Information about each data set can be found at their individual source location:

Current public data for the FTIR, lidar, MWR, and Umkehr can be found at NDACC: https://ndacc.larc.nasa.gov/data. The FTIR data will be updated here with the new strategy explained in the paper.

NOAA Dobson Total Column Ozone measurements can also be found on the NOAA GML FTP website here: https://gml.noaa.gov/aftp/ozwv/Dobson/, at WOUDC: https://woudc.org/. Likewise, the Monthly Mean optimized/homogenized Umkher profiles can be found on the NOAA GML FTP here: https://gml.noaa.gov/aftp/ozwv/Dobson/AC4/Umkehr/Optimized/.

Ozonesondes: The homogenized Lauder ozonesonde time series are available at the HEGIFTOM ftp-server https://hegiftom.meteo.be/datasets/ozonesondes.

**Table A1.** Changes from the FTIR V08 retrieval strategy for ozone to the new strategy. The regularization strength $\alpha$ is specific to the Lauder measurements.

|  | V08 strategy | new strategy |
|---|---|---|
| Spectroscopy | HITRAN2008 | HITRAN2020 |
| Microwindows | 1000-1005 cm$^{-1}$ | 991.25-993.8 cm$^{-1}$ |
|  |  | 1001.47-1003.04 cm$^{-1}$ |
|  |  | 1005.0-1006.9 cm$^{-1}$ |
|  |  | 1007.35-1009.0 cm$^{-1}$ |
| Regularization | OEM | Tikhonov with $\alpha = 1000$ |
| A priori | WACCM v6 | WACCM IRWG |

## Appendix A: Influence of FTIR retrieval setup

### A1 New FTIR retrieval strategy

The FTIR ozone retrievals used in this study employ an improved retrieval strategy compared to Vigouroux et al. (2008) (V08) that has been tested at several NDACC sites and became recently the recommended strategy for the IRWG. The most notable changes to the strategy are listed in Table A1. Firstly, necessary input for atmospheric retrievals is a spectroscopic line list including information on the wavenumber of spectral lines and their line strengths for many molecules. Such information is contained in the spectroscopic database of HITRAN (HIgh resolution TRANsmission), where the IRWG at present uses the HITRAN2008 database (Rothman et al., 2009). In the FTIR retrievals used in this study, we have updated the spectroscopic line list to use the latest HITRAN2020 database (Gordon et al., 2022). We find that a result of this change in spectroscopy is generally that retrieved ozone columns are reduced by $2 - 3\%$. Secondly, the spectral range that is fitted in the retrieval of ozone (the microwindows) spanned from 1000 to 1005 cm$^{-1}$ in the V08 strategy. In the new strategy we use a combination of 4 smaller microwindows, which are chosen such that they avoid strong interference from water vapor lines (García et al., 2022; Schneider and Hase, 2008). Thirdly, the choice of constraints to solve the inverse problem within the retrieval method is chosen as a specific regularization matrix. This matrix can be chosen through the Optimal Estimation Method (OEM, Rodgers, 2000) where the matrix is the inverse of the a priori covariance, or through a so-called smoothing constraint such as the Tikhonov regularization (Tikhonov, 1963). While the first option in theory provides a better regularization from climatological constraints, in practice (such as in the V08 FTIR retrieval strategy) usually a simplified matrix is adopted to represent the variability on the retrieved profile. In the new FTIR strategy we opt to use the Tikhonov regularization, where the strength of the variability has to be determined by considering the DOFS and the retrieval noise error (Steck, 2002). One important difference with the other regularization method is that only the shape of the profile is constrained within the Tikhonov method and not the absolute value, which reduces trend bias due to a statistic a priori. Lastly, since prior information is important and influential for atmospheric retrievals, we also need to consider the choice of the a priori. This a priori information comes from the Whole Atmosphere Community Climate Model (WACCM, Marsh et al., 2013), where we now adopt a different version named 'IRWG' (Keeble et al., 2021) in stead of version 6 (Gettelman et al., 2019). However, we find no significant effect in comparison to retrievals performed using the V08 WACCM v6 a priori, but still adopt this change for consistency with the retrieval of other molecules that are targeted by the IRWG which do have significantly different a priori profiles.

### A2 Comparison to the FTIR V08 strategy

To elaborate on the differences between the FTIR V08 retrieval strategy of Vigouroux et al. (2008) and the new strategy explained in Section 2.1, we showcase here the intercomparison study from above performed with both retrieval strategies. First of all, we consider the bias of the total column of FTIR with Umkehr. The bias of TCO retrieved with the FTIR V08 strategy is $-5.7\%$ with Umkehr, in relative difference as before. This value is higher in absolute value than that found using the new retrieval strategy, where we find a bias of $-2.9\%$ with Umkehr. A similar reduction of $1 - 3\%$ in the bias is also found in most partial columns comparing to the other measurements. Additionally, the MAD$_s$ value for the differences in total column between FTIR and Umkehr are also seen to be reduced from $2.0\%$ using the V08 trategy to $1.7\%$ using the new strategy. This shift in lower ozone columns for the new FTIR strategy is mostly due to the change in spectroscopy from HITRAN2008 to HITRAN2020. Namely, when performing retrievals using the V08 strategy only changing the spectroscopic data, the columns generally are seen to reduce by 2-3%, which matches the changes in the biases with respect to the other measurements and brings them all in decent agreement with one another considering the instrument uncertainties.

Second of all, the new FTIR retrieval strategy also causes differences in the drifts with respect to the other ground-based instruments. For comparison with earlier results, the drifts derived through the same intercomparison method, but now using the FTIR V08 strategy, are shown in Table A2. Overall, the change from the V08 to the new strategy improves most of the drifts and some even change from being significant to being non-significant within the trend error. In the troposphere we find that, while the ozonesonde data does have a drift with respect to FTIR V08 observations, this drift is no longer present using the new FTIR strategy. In the lower stratosphere, while the drifts do change slightly in value, the conclusions remain the same. In the middle stratosphere the drift with Umkehr is reduced to become non-significant with

**Table A2.** Drifts with respect to the FTIR V08 retrieval strategy for each of the measurements in %/decade for every partial column where they have data available. The drifts that are significantly different from zero (considering $U_{\mathrm{drift}}$) are represented in boldface.

|        | 0.5-11 km | 14-21 km | 21-29 km | 29-42 km |
|--------|-----------|----------|----------|----------|
| Umkehr | $-0.8 \pm 4.7$ | $\mathbf{2.7 \pm 1.1}$ | $\mathbf{1.9 \pm 1.8}$ | $\mathbf{-3.5 \pm 1.1}$ |
| Sonde  | $\mathbf{-3.2 \pm 2.1}$ | $-0.7 \pm 1.7$ | $0.5 \pm 1.5$ | – |
| Lidar  | – | $0.8 \pm 1.9$ | $\mathbf{3.0 \pm 1.0}$ | $\mathbf{-5.2 \pm 1.5}$ |
| MWR    | – | – | $\mathbf{2.5 \pm 2.4}$ | $-1.5 \pm 1.9$ |

the new FTIR data. However, one change to remark here is that, while all drifts in the middle stratosphere are reduced, the drift of MWR with the new FTIR actually increases. This is mainly due to the fact that the time span of the MWR data only lasts until October 2016. The FTIR trend computed on this shorter time series is actually seen to increase when changing from the V08 to the new strategy, which is the reverse of what happens when the full time series is considered. Potentially then, should the missing 5 years have been included in the MWR observations, following the logic of the other measurements, the drift would have decreased when using the new FTIR and may even no longer be significant. Lastly, in the upper stratosphere, the conclusions remain unchanged when adopting either the V08 or new FTIR retrieval strategy. Both drifts of Umkehr and lidar with the new FTIR are lower in absolute value than with the V08 FTIR data.

Previously, we saw that the change in spectroscopy affects the retrieved ozone columns for FTIR, reducing the bias with all other measurements. This effect, however, is the same over the full data set and thus will not alter the derived drifts. The relevant change in the retrieval strategy here is that the regularization in the new strategy uses Tikhonov regularization instead of optimal estimation or that we use a higher regularization strength than in the V08 strategy bringing down the DOFS. The change in regularization slightly reduces both the positive and the negative trends of FTIR in all partial columns. The consequence we see here is that, when using Tikhonov regularization, the drifts generally improve, and we even find that there is no significant drift between FTIR and ozonesonde for any of the partial columns.

## Appendix B:  Choice of time window

The time window where two measurements are used to be compared to one another should not be too large, as to avoid comparing different times of day where ozone could naturally change within the diurnal cycle or by comparing completely different air masses. In part, this is determined by analyzing for which time window there is the lowest bias and scatter between the measurements which are all listed in Table B1. The choice is also made such that there are plenty of comparisons available to sample the full time coverage, be-

cause sampling is crucial in long-term intercomparison studies, and by simultaneously checking the behavior of the drift. The Umkehr observations are only made at sunrise and sunset, so we find that 6 hours is the ideal time window to still have plenty of comparison pairs over the full time series with the best values of M and $\mathrm{MAD}_s$ over the partial columns as seen in Table B1. For the same reason, the time window of 6 hours is selected for the comparison to the ozonesonde observations. Because these measurements are not very frequent, we find that we need a large enough window to get a dense enough sampling over the time series. The microwave radiometer measurements are taken a lot more frequently, so a smaller time window of 3 hours here is sufficiently large enough to construct many comparison pairs. The time window of 1h does not really improve the bias or scatter and furthermore makes the bias more negative. It is important not to take a too large time window, because of the diurnal variation and short term variability as discussed in Sauvageat et al. (2023). They note a high variation, especially at the stratopause. We therefore check the extent of the variability in the upper stratosphere for the MWR measurements. We find the short term variability to be at most 18 % and assess from this that a 3 hour time window is sufficiently small (with a mean value of 1.1 %) such that the diurnal variation will not impact the intercomparison study. Most problematic to construct the comparison pairs however are the lidar observations. These are taken exclusively at night, while FTIR measurements are taken exclusively during the day, because direct sunlight is necessary. A time window of 12 hours is necessary to reach a decent sampling over the available time series.

## Appendix C:  Effects of smoothing

We explore the difference between the comparisons performed with and without smoothing the high-resolution profile. Both results are shown in Figure C1, where the bias is shown over the full ozone profile together with the $\mathrm{MAD}_s$ in the shaded areas. When we perform the intercomparion without smoothing, a lot more oscillation is seen in the relative difference of the profiles. This is especially pronounced for the sonde data, which (as we can also see in Figure 3) shows a lot sharper oscillation than the FTIR profile. These are actual profile measurements from ozonesondes, so direct comparison with FTIR does make sense and limits any influence of trends in the FTIR averaging kernel on the ozonesonde trends (García et al., 2012). However, because the vertical resolution of FTIR is much smaller, these high spatial oscillations can never be observed so it makes sense to adjust the ozonesonde profile to incorporate the same vertical information. We see that this results in overall a better comparison of the profile to FTIR. Similar results, although less pronounced, can be seen for the comparison with the other measurements. When we divide into partial columns, this ef-

**Table B1.** Bias and MAD$_s$ values for several different time windows where comparison pairs between FTIR and other measurements are constructed and the number of coincidences found in those windows. Lidar is omitted here because a window of less than 12h would leave insufficient temporal sampling.

| | 0.5-11 km M [%], MAD$_s$ [%] | 14-21 km M [%], MAD$_s$ [%] | 21-29 km M [%], MAD$_s$ [%] | 29-42 km M [%], MAD$_s$ [%] | Number of coincidences |
|---|---|---|---|---|---|
| Umkehr 3h | 9.7, 16.7 | -4.2, 3.5 | -7.1, 4.6 | - 1.1, 4.7 | 1625 |
| Umkehr 6h | 8.8, 17.1 | -4.3, 3.5 | -7.4, 4.6 | -1.0, 4.7 | 3526 |
| Umkehr 12h | 8.7, 17.5 | -4.5, 3.8 | -7.4, 4.6 | -0.9, 4.7 | 3808 |
| Sonde 3h | -2.9, 5.1 | -6.3, 3.8 | -6.7, 3.5 | | 596 |
| Sonde 6h | -2.6, 5.4 | -6.2, 4.3 | -6.8, 3.4 | | 929 |
| Sonde 12h | -2.5, 6.0 | -6.8, 4.5 | -6.8, 3.5 | | 1130 |
| MWR 1h | | | -4.9, 5.0 | 0.3, 6.3 | 1029 |
| MWR 3h | | | -5.4, 5.1 | -0.1, 6.1 | 2176 |
| MWR 6h | | | -5.0, 5.0 | -0.2, 6.0 | 2772 |

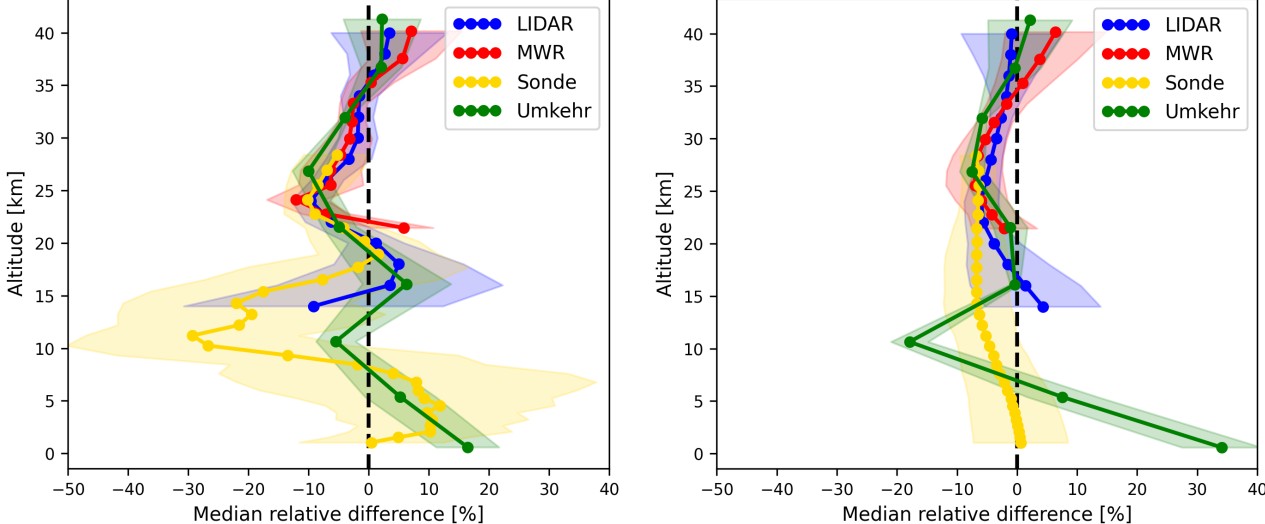

**Figure C1.** Median bias of the profile when comparing FTIR to lidar, MWR, ozonesonde, and Umkehr together with the MAD$_s$ shown as the colored, shaded areas. The left panel shows the differences between the profiles when not applying the smoothing step, while the right panel does smooth the higher resolution profile with the averaging kernel of the lower resolution measurement.

fect of smoothing on the derived results should not be too big, because we chose the columns such that we have around one DOFS for FTIR. The only exception is in comparing FTIR to Umkehr in the partial columns where the Umkehr DOFS are less than one, where we now use the Umkehr averaging kernels instead. We see for example that the only place where the profile comparison seems to get worse when applying smoothing, is for the Umkehr comparison in the troposphere (where Umkehr reaches a DOFS of 0.5).

To analyze the effects of smoothing on the drift between the instruments, Table C1 shows these values with their uncertainties when smoothing is not applied during the intercomparison. No big differences are seen and the drifts that are deemed significant considering the uncertainty are the same both with and without smoothing. For the ozonesonde

comparison, the drift reduces slightly in all partial columns, strengthening the use of a smoothed profile in this comparison. For the other measurements, however, the changes are not as consistent. Both lidar and Umkehr have partial columns where the drift improves after smoothing the profile as well as worsens. For the MWR comparison we see a small increase in the drift in both partial columns. Because the changes are not very large, not changing conclusions about significance within the drift uncertainty, and generally the bias and drifts seem to improve (especially with ozonesonde data) we have chosen to work with a smoothed profile in the intercomparison study.

**Table C1.** Drifts with respect to FTIR for each of the measurements in %/decade for every partial column where they have data available. The values for the drifts without performing the smoothing step in the intercomparison are shown. The drifts that are significantly different from zero (considering $U_{\mathrm{drift}}$) are represented in boldface.

| Drifts, not smooth [%/decade] | Troposphere 0.5-11 km | Lower stratosphere 14-22 km | Middle stratosphere 22-29 km | Upper stratosphere 29-42 km |
| --- | --- | --- | --- | --- |
| Umkehr | 0.1±4.6 | **2.7 ± 1.4** | 1.1±1.4 | **-2.8 ± 1.1** |
| Sonde | 0.2 ± 2.5 | -1.1 ± 1.4 | 0.2 ± 1.6 | – |
| Lidar | – | -1.1 ± 1.9 | **2.5 ± 0.9** | **-3.1 ± 1.2** |
| MWR | – | – | **3.0 ± 2.0** | -1.4 ± 2.0 |

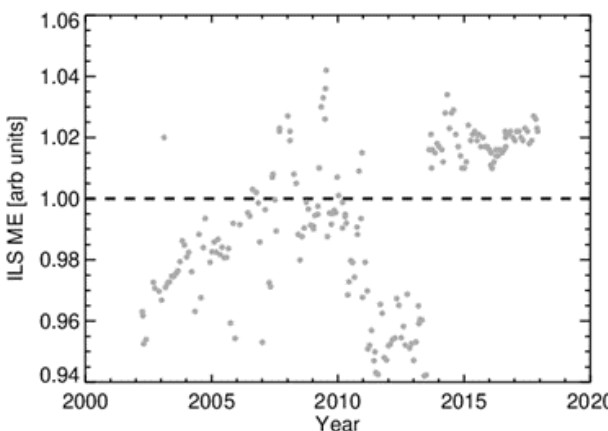

**Figure D1.** Lauder Bruker 120HR ILS modulation efficiency at 148 cm optical path difference

## Appendix D: Calibration of FTIR and Dobson

### D1  FTIR

FTIR instrument performance (accuracy, precision, and stability) is routinely assessed via monthly cell measurements. The closed cell contains a low-pressure gas. The cells employed to date have contained pure mixtures of $\sim$0.2 hPa HBr or $N_2O$ (Coffey et al., 1998; Hase, 2012). Analysis of the cell spectra is via the code LINEFIT (Hase et al., 1999) which is used to retrieve the instrument line shape (ILS) and cell total column abundance along with other instrument diagnostics. Multiple measurements over time gives instrument stability and precision. Bias (accuracy) is deduced from the difference in the retrieved total column to the reference total column amount.

At Lauder, monthly HBr cell tests were made on the Bruker 120HR from 2002 to 2018. For the Bruker 125HR cell tests (2018 to present) a $N_2O$ cell was used. Figure D1 displays the modulation efficiency (ME: measured ILS relative to theoretical ILS) for the Lauder 120HR. Values with $\sim$4% of unity indicate a well aligned instrument (pers. comms Frank Hase). Between 2002 and 2018 there were over 500 instru-

ment events. Long term observations were constantly interspersed with tests and component failures and changes. Each event has potential to influence instrument performance. Two prominent examples of these are large step changes in the ILS ME in December 2010 and August 2013 (see Fig. D1). The cause of these were major optical realignments. Such ME step changes are not uncommon within the NDACC FTIR community (e.g. García et al., 2021) and the effect (of ME changes) on total (and partial column) retrievals have been investigated recently by García et al. (2021) and Sun et al. (2018). The effect of the December 2010 and August 2013 ME changes are not discernible in instrument comparisons.

The statistically inferred discontinuity identified in May 2018 (Figure E1) is related to the change of the instrumentation from the Bruker 120HR to Bruker 125HR. The effect of this instrument transition on total column abundances on multiple species has been documented in Smale (2019).

### D2  Dobson

The Lauder Dobson (#72) undergoes routine monthly calibrations using a mercury lamp and quartz halogen lamps to monitor variations in wavelength and instrument stability. As part of NDACC protocols, every 5 years the Lauder Dobson is transported to Melbourne to be intercompared with the Regional Standard Dobson (#105), maintained and operated by the Australian Bureau of Meteorology. Dobson intercomparisons since 2000 were performed in 2006, 2012, 2017, and 2022.

An issue due to rain damage was identified at the 2012 calibration campaign and data from 2006-2012 were reprocessed. A full description of this is given in Evans et al. (2017).

NOAA installed the automated Dobson (#72) at Lauder in 1987. The automation system was updated to the WinDobson system in 2012. Within the NOAA Dobson network, semi-automation refers to capturing R-dial values with an encoder and assisting the measurement procedure with a computer. The Lauder Dobson automation also includes table rotation and zenith hatch control for unattended zenith observations; direct Sun observations require an operator to open the dome

and position the sun director.

At the 2012 intercomparison (IC2012), a comparison was made with the Secondary Reference Instrument (#65). The final error with the World Standard Dobson (#83) was adjusted
to zero. Similar adjustments were made in the 2006 calibration as well. During the last two intercomparisons (IC2017 and IC2022), the World Standard Dobson (#83), maintained and operated by NOAA at Boulder/Mauna Loa, also participated in the Melbourne campaign, so the Lauder Dobson is
directly traceable to the World Standard Dobson (#83). From the IC2022 report, the Lauder NAD ADD value (0.66) implied an average -0.8% error in the calculated ozone value, over the range Mu=1.1 to 2.5, for Total Ozone = 300 DU. Generally, an error of ±1% or less is within the acceptable
range, therefore no recalculations were applied to Lauder Dobson data since IC2017.

## Appendix E: Effect of discontinuities

In the drift analysis we found that temporal sampling has an important influence on the calculated drift between two mea-
20 surements. Here we additionally study the influence of potential steps or discontinuities (change point), which for example could arise from physical changes to the measurement instrument. In order to find a change point in the time series of relative differences, we use the Lanzante change-point detec-
25 tion algorithm (Lanzante, 1996) as is similarly done in (García et al., 2014). The Lanzante's algorithm iteratively finds a change point from summing the ranks of the time series from the beginning to each point in the series. Afterwards the series is adjusted using the median of the subseries enclosed by
30 the currently found change points. This method is repeated on the adjusted time series until the found change point is statistically insignificant (p-value<0.05). This method of change-point detection is purely statistical, hopefully aligning with changes to the retrieval strategy or changes to the
35 instrument itself as a cause of the found step.

This method is applied to the relative differences in total and partial columns where we have calculated the drifts in order to find any recurring discontinuities that could point to steps in any of the measurement time series. Multiple
change-points over the different intercomparisons are identified using the Lanzante method. One of these steps in the middle of 2018 seems to reappear for multiple measurement comparisons, pointing to a step in the FTIR data (being the reference measurement). To analyze this discontinuity, we
show the time series of FTIR DOFS in Figure E1 and again apply the Lanzante change-point detection. Because seasonality is present (which was not the case for the relative differences), we first apply a seasonal fitting to the time series and subtract it. We fit the DOFS $Y(t)$ using

$$Y(t) = A_0 + A_1 t + A_2 \sin(2\pi t) + A_3 \cos(2\pi t)$$
$$+ A_4 \sin(4\pi t) + A_5 \cos(4\pi t)$$

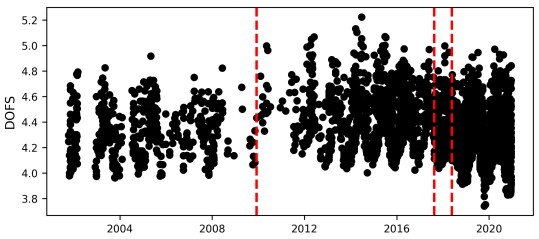

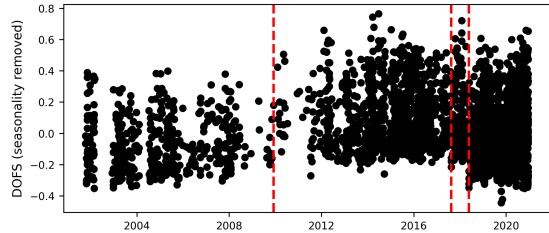

**Figure E1.** DOFS time series before removing seasonality (upper figure) for the FTIR total column measurements and when removing seasonality as described in the text (lower figure). The statistically significant change points identified by the Lanzante approach are shown in a red, dashed line. The most prominent change point falls around May 2018, with subsequent points found around December 2009 and August 2017.

with $A_0$ the intercept, $A_1$ the slope and the other terms are the seasonality. From the de-seasoned DOFS we find again the same change point we found for multiple relative differences in May 2018. Since the retrieval strategy is the same over the
55 full time series, we search for the cause of the steps in the instrument itself. From the FTIR instrument logs, we find that on 10 May 2018 there was a 'major alignment' which is the most likely cause for the step in the FTIR data.

Once this change point is found, we can recalculate the
60 drift for each time series of relative differences both before and after this point to find the influence of the discontinuity on the overall drift. Because the step is towards the end of the time series and there are not a lot of data points, the drift to the right of this step has a very high uncertainty. The drift
on the left of the step is always similar to the overall drift from Table 4. This means that, for this time series, the step in the FTIR data set caused by the major alignment does not significantly affect the overall drift reported in this study.

For the other two change points, we cannot identify a clear
instrumental reason. Similar to the previous change point, if we calculate the drift left and right of the discontinuities, we find a drifts on the same order as for the total time series, but with larger uncertainties. Such a change-point analysis requires more attention and future work to identify for each
instrument the discontinuities and relate them to changes to the instrument or data-processing techniques, but this falls beyond the scope of this paper.

*Author contributions.* RB has carried out the majority of the inter-comparison study and writing the major part of the text. This was done in close collaboration with CV who supplied the trend results and additionally wrote significant parts of the text. The code to perform the intercomparisons was provided by BL. PE, IP, GN, RVM, AG, and RQ provided text concerning their instrument expertise. Additionally, DS and RQ provided the full text and figures for Appendix D. RB, CV, OG, JH, GN, IO, IP, RQ, DS, RVM, and MDM all participated in the discussion and provided comments on the text. DS, PE, AG, KM, MK, GN, IP, DP, RQ, JR, HS, PS, and RVM all contributed in supplying the data sets where RVM additionally created a preliminary ozonesonde data set with the TRCC method.

*Competing interests.* At least one of the (co-)authors is a member of the editorial board of AMT.

*Acknowledgements.* The work of RB has been supported by BIRA-IASB and is performed within the context of LOTUS and TOAR-II.

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
