# Peer review of "Intercomparison of long-term ground-based measurements of total, tropospheric and stratospheric ozone at Lauder, New Zealand"

_EGUsphere, 2023_

## Referee Comment (RC2)

Review of Björklund, Vigouroux, Effertz et al: Intercomparison of long-term ground-based measurements of tropospheric and stratospheric ozone at Lauder, New Zealand (45S)                                          12 March 2024

**Summary of Paper**

There are five ground-based (GB) instruments at Lauder, New Zealand, that have measured total column ozone and/or partial columns throughout the interval 2000 to 2022.  They do not appear to all give the same values in total ozone or in various segments: troposphere, lower stratosphere, middle stratosphere, upper stratosphere. Accordingly, computed trends over the 23-year period differ, especially in the lower stratosphere (LS) where LOTUS has concentrated its efforts.  A major goal of this paper, as expressed in the Abstract, is to determine why LOTUS trends are not similar among the techniques. The second goal is to determine "quality and relevance" for TOAR II trends, two criteria that are not well-defined.

This paper makes comparisons of the ozone amounts systematically within the segments, using FTIR as the primary reference. Of the four independent measurement types considered, three (Lidar, Microwave, Umkehr) all display significant drift relative to FTIR in one or more stratospheric segments; ozonesondes do not (Table C1). Certain discontinuities near the end of the record contribute to these drifts and the divergence of trends (Appendix D). A reprocessing (modified FTIR retrieval) improves some of the drifts.  In summary, the paper contains worthy analyses, carefully carried out.

However, there are two reasons why the paper is not ready for publication. First, after all the tables and analyses, the paper does not come back to clear answers to guide how past LOTUS results concerning Lauder can be updated. Nor does the paper provide recommendations for TOAR II activities on how to use the findings in trends analyses. For example, should one try to merge the various datasets for tropospheric ozone analysis? Why or why not?  If so, how would that be done?  The paper needs to be re-outlined and clear conclusions on how, if and why each of the 5 datasets can be used in ongoing LOTUS and TOAR II analyses.

Second, there are more fundamental questions about the Lauder datasets relevant to LOTUS and TOAR II.  Here are several:

1. In the TOAR II HEGIFTOM activity, presumably the FTIR, Umkehr, sonde records have been homogenized. The paper gives no information about the data version, archive, etc, for each of these data sets.  Are these the HEGIFTOM files at the RMI ftp repository?  The customary doi information is lacking

2. With respect to the ozonesonde data in particular, papers by Stauffer et al (2020; 2022) and updates (through 2021, see Figures below) find total ozone column and stratospheric ozone in particular, suffered the "Ensci dropoff" artifact at Lauder. The upper figure is a satellite comparison – Aura MLS for stratosphere, OMI, OMPS and European TCO comparisons. The lower Figure is based on the Lauder Dobson as archived at WOUDC, Dobson presumably the source of the Umkehr data. Have ozonesonde dropoffs been corrected in the HEGIFTOM files? The wording about the version of sonde data (page 9 of

the manuscript) is vague. Reprocessing via the Smit method, even the WMO/GAW, 2021, Report, as referenced (line 217 to 220) do not give a procedure for correcting for the dropoff. Was the process of Nakano & Fujimora, *AMT,* 2023) to correct the dropoff applied to the Lauder record? "Claim to be removed" is your wording – what does that mean? If the dropoff has been fixed, it would be good to have a supplementary figure showing that.

3. If the dropoff has not been corrected, the authors need to implement the Nakano and Fujimora (2023) procedures; ideally the new reprocessing by Smit et al (*AMT,* 2023) would lead to an even more accurate, referenced result.  For LOTUS applications the FTIR-referenced comparisons make sense but for the TOAR II application in the troposphere, the optimized sonde data should also be used as the reference.

4. In the case of TOAR II/HEGIFTOM, calculations for 2000-2022 trends being prepared for publication (VanMalderen et al) show the following. Note that trends for the HEGIFTOM ozonesonde data at Lauder (surface to 300hPa) and trends for Umkehr and FTIR at Lauder diverge somewhat as shown below. Graphs of this information were presented to the HEGIFTOM Teams meeting of 7 March.  (Based on calculations from NOAA and GSFC)

| 2000-2022 Trends | Surface to 300 hPa | Not rounded to sig fig | | QR L1 (ppbv/dec) | QRL3 (ppbv/dec) | MLR L3 (ppbv/dec) |
|---|---|---|---|---|---|---|
| Lauder | O3S | -45 | 169.68 | 0.134324342 | 0.01106383 | 0.133214349 |
| | FTIR | -45.04 | 169.68 | 1.544135587 | 1.638209739 | 1.673699546 |
| | Umkehr | -45.04 | 169.68 | 0.358046 | 0.377753 | 0.579331805 |

It is assumed that the data used in the above Table are the same as Björklund et al are using but more details are required in Section 2.  *RELATED COMMENT IN RESPONSE TO OWEN COOPER COMMENT ON THIS PAPER.(see* https://doi.org/10.5194/egusphere-2023-2668-CC1). The table above shows that there is sufficient variation in the surface to 300 hPa trends for sonde, Umkehr and FTIR that "averaging the data" (as Cooper recommends) or averaging the trends is not justified. The current manuscript and the trends analyses show that, in a revised manuscript, more analyses need to be carried out, with careful uncertainty comparisons, on the FTIR, Umkehr and sondes before merging of data can be considered, as suggested by Cooper. It is particularly important that uncertainties for the 5 different instruments being considered are compared.  Note that **Figure 1** in the manuscript suggests that FTIR and sonde TCO had some declines, albeit not montonic or identical, after 2014.

A further comment on the Cooper et al Comment on this paper. Reference is made to the Pope et al RAL paper: Atmos. Chem. Phys., 23, 14933–14947, 2023 https://doi.org/10.5194/acp-23-14933-2023. That paper was accepted prior to the reprocessing of OMI (2014-2021) data that displayed a drift artifact in total ozone. The latter issue is discussed in with corrected data by co-author Ziemke in Gaudel et al:

https://egusphere.copernicus.org/preprints/2024/egusphere-2023-3095/. The Pope et al., RAL product overestimates tropospheric ozone trends.

In summary, the paper in its present form should not be published. In a revision the authors need to:
  (1) clarify the source of their data – the customary DOIs and references on the datasets are absent.
  (2) If the sonde data are not corrected for an artifact stratospheric ozone loss after 2014, that needs to be done before re-analyzing drifts. Intrinsically, the sonde data are more accurate than FTIR in the troposphere and possibly in the lowest and mid-stratosphere. Drifts in FTIR for those segments *relative to corrected sonde data* should be carried out and discussed for the troposphere, lower and mid-stratosphere.
  (3) Most important, please think through and describe clearly the significance of the new results for LOTUS and TOAR II/HEGIFTOM. The paper currently presents interesting technical details but does not relate a clear scientific story of interest to the TOAR II community.

Lesser comments:
  (1) Section 2.5. Note that the sonde instrument type and solution used at Lauder should be added. On line 214, end of sentence, the following reference for the variations in types of instrument and solutions should be inserted.
      H. G. J. Smit, A. M. Thompson and ASOPOS, Ozonesonde Measurement Principles and Best Operational Practices, ASOPOS (Assessment of Standard Operating Procedures for Ozonesondes) 2.0, 165 pp., WMO/GAW/IO3C/NDACC/GRUAN, WMO/GAW Report 268, Geneva. (Online at https://library.wmo.int/index.php?lvl=notice_display&id=21986#.YaFNSbpOlc8). Alternatively this can be called WMO/GAW 2021 but the citation is missing from the Reference list at the end of the manuscript

  (2) The authors have done a fine job in English but there remain many English errors. Please ask authors 3, 5 or 6, as appropriate to review and correct them.

  (3) The Stauffer references for figures below:
      Stauffer, R. M., A. M. Thompson, D. E. Kollonige, J. C. Witte, D. W. Tarasick, J. M. Davies, H. Vömel, G. A. Morris, R. Van Malderen, B. J. Johnson, R. R. Querel, H. B. Selkirk, R. Stübi, H. G. J. Smit, A post-2013 drop-off in total ozone at third of global ozonesonde stations: ECC Instrument artifacts?, *Geophys. Res. Lett.,* doi: 10.1029/2019/GL086791, 2020.
      Stauffer, R. M., A. M. Thompson, D. E. Kollonige, D. W. Tarasick, R. Van Malderen, H. G. J. Smit, H. Vömel, G. A. Morris, B. J. Johnson, P. D. Cullis, R. Stübi, J. Davies, M. M. Yan, An examination of the recent stability of ozonesonde global network data, Earth Space. Sci., https://doi.org/10.1029/2022EA002459, 2022.

Figure showing ozonesonde 'dropoff' for TCO and stratospheric ozone in the Lauder record (Stauffer et al., 2020; Stauffer et al, 2022 & updates).  Files were downloaded from RMI ftp site, 2021. The lower comparison is sonde TCO vs TCO from the co-located Dobson.

---

## Community Comment (CC1)

Comments by Owen R. Cooper (TOAR Scientific Coordinator of the Community Special Issue) on:

**Intercomparison of long-term ground-based measurements of tropospheric and stratospheric ozone at Lauder, New Zealand (45S)**

Robin Björklund (corresponding author), Corinne Vigouroux, Peter Effertz, Omaira Garcia, Alex Geddes, James Hannigan, Koji Miyagawa, Michael Kotkamp, Bavo Langerock, Gerald Nedoluha, Ivan Ortega, Irina Petropavlovskikh, Deniz Poyraz, Richard Querel, John Robinson, Hisako Shiona, Dan Smale, Penny Smale, Roeland Van Malderen, and Martine De Mazière

This manuscript was submitted to AMT as part of the TOAR-II Community Special Issue
https://doi.org/10.5194/egusphere-2023-2668, 2023

This review is by Owen Cooper, TOAR Scientific Coordinator of the TOAR-II Community Special Issue. I, or a member of the TOAR-II Steering Committee, will post comments on all papers submitted to the TOAR-II Community Special Issue, which is an inter-journal special issue accommodating submissions to six Copernicus journals:  ACP (lead journal), AMT, GMD, ESSD, ASCMO and BG. The primary purpose of these reviews is to identify any discrepancies across the TOAR-II submissions, and to allow the author teams time to address the discrepancies.  Additional comments may be included with the reviews. While O. Cooper and members of the TOAR Steering Committee may post open comments on papers submitted to the TOAR-II Community Special Issue, they are not involved with the decision to accept or reject a paper for publication, which is entirely handled by the journal's editorial team.

**General comments:**

This paper presents a thorough multi-instrument comparison of total column ozone, stratospheric ozone and tropospheric ozone above Lauder, New Zealand focusing on the period 2000-2022. My comments will focus on just the tropospheric portion of the analysis.  While typical measurement uncertainties are considered, the issue of low sampling frequency also needs to be addressed. As described below, low sampling frequency is a major challenge to accurate trend quantification.  However, because Lauder has high quality observations from several instruments this analysis provides an excellent opportunity to calculate the tropospheric ozone trend by merging all available data. The method of calculating ozone trends and anomalies based on the merging of data from nearby stations has been done several times before (Cooper et al., 2010; Tarasick et al., 2010; Gaudel et al., 2018; Steinbrecht et al., 2021; Chang et al., 2022), but this paper could produce the first (as far as I am aware) ozone trend analysis that merges three data sets at a single location.  By merging the data sets and greatly increasing the sample size the authors may succeed in reducing the uncertainty on the long-term trend, which could serve as a model for future TOAR-II trend studies.

Between 1988 and 2020 five papers appeared in the peer-reviewed literature showing that a low sampling rate of just once per week (based on ozonesondes) either failed to produce an accurate monthly/seasonal mean (Logan 1999; Saunois et al., 2012), or failed to produce an accurate trend (Prinn, 1988; Cooper et al., 2010; Chang et al., 2020).  While these papers have been around for a long time, the sampling issue is often overlooked. Chang et al. (2023) have written a new paper on ozone sampling rates and they offer some suggestions for improved trend detection when faced with low sampling rates (this paper will soon appear as a submission to the TOAR-II Special Issue).  The simplest

solution is to boost the sampling rate, and in the case of Björklund et al. (2023) that can be achieved by merging the ozonesonde, FTIR and Umkehr records at Lauder. According to Table 1 the merged data set could have more than 10 profiles per week, although probably only for the most recent years, as earlier years had gaps, and Figure 1 seems to indicate lower sampling rates before 2012. Table 5 shows that the individual trends from ozonesondes, Umkehr and FTIR have similar magnitudes but all have relatively wide 95% confidence intervals. A merged data set may produce a trend that is more accurate and with lower uncertainty. This finding would also be relevant to the first paper published in the TOAR-II Community Special Issue. Pope et al. (2023) developed a new satellite ozone product that indicates broad ozone increases across much of the globe from 1996 to 2017. The product shows a weak ozone increase above New Zealand, while the trend based on the sparse Lauder ozonesonde record shows no trend. Would a merged ozone time series based on ozonesondes, FTIR and Umkehr reconcile this discrepancy?

**Comments regarding TOAR-II guidelines:**

TOAR-II has produced two guidance documents to help authors develop their manuscripts so that results can be consistently compared across the wide range of studies that will be written for the TOAR-II Community Special Issue. Both guidance documents can be found on the TOAR-II webpage: https://igacproject.org/activities/TOAR/TOAR-II

*The TOAR-II Community Special Issue Guidelines*: In the spirit of collaboration and to allow TOAR-II findings to be directly comparable across publications, the TOAR-II Steering Committee has issued this set of guidelines regarding style, units, plotting scales, regional and tropospheric column comparisons, and tropopause definitions.

*The TOAR-II Recommendations for Statistical Analyses*: The aim of this guidance note is to provide recommendations on best statistical practices and to ensure consistent communication of statistical analysis and associated uncertainty across TOAR publications. The scope includes approaches for reporting trends, a discussion of strengths and weaknesses of commonly used techniques, and calibrated language for the communication of uncertainty. Table 3 of the TOAR-II statistical guidelines provides calibrated language for describing trends and uncertainty, similar to the approach of IPCC, which allows trends to be discussed without having to use the problematic expression, "statistically significant".

It would be helpful for TOAR-II if the submitted paper can describe the trend detection method that was applied (e.g. linear least squares, quantile regression, multiple linear regression), report all trends with 95% confidence intervals and p-values, and avoid using the expression, "statistically significant". Based on the highly influential paper by Wasserstein et al. (2019), TOAR first abandoned the expression "statistically significant" with the *TOAR-Observations* paper (Tarasick and Galbally et al., 2019), and we now ask the author teams to describe their confidence in a trend using the calibrated language in the statistical guidelines. In addition, to facilitate trend comparisons across TOAR-II papers, trends should also be reported in units of nmol mol$^{-1}$ per decade (it's fine to also report trends in units of percent per decade, as currently shown in the submitted manuscript).

**Additional Comments:**

References to the WMO 2018 ozone assessment should be updated with the 2022 edition.

Reference to the National Research Council 1991 report needs to be updated with a modern review of the impact of ozone on human health.

When providing a summary of global tropospheric ozone trends, the TOAR paper by Gaudel et al. (2018) has been superseded by IPCC AR6, which assessed an increase of the tropospheric ozone burden since the 1990s in both the tropics and northern mid-latitudes (Gulev et al., 2021; Szopa et al., 2021).

**References**

Chang, K.-L., O. R. Cooper, A. Gaudel, I. Petropavlovskikh and V. Thouret (2020), Statistical regularization for trend detection: An integrated approach for detecting long-term trends from sparse tropospheric ozone profiles, Atmos. Chem. Phys., 20, 9915–9938, https://doi.org/10.5194/acp-20-9915-2020

Chang, K.-L., O. R. Cooper, A. Gaudel, M. Allaart, G. Ancellet, H. Clark, S. Godin-Beekmann, T. Leblanc, R. Van Malderen, P. Nédélec, I. Petropavlovskikh, W. Steinbrecht, R. Stübi, D. W. Tarasick, C. Torres (2022), Impact of the COVID-19 economic downturn on tropospheric ozone trends: an uncertainty weighted data synthesis for quantifying regional anomalies above western North America and Europe, AGU Advances, 3, e2021AV000542. https://doi.org/10.1029/2021AV000542

Chang, K.-L., O. R. Cooper, A. Gaudel, B. C. McDonald, I. Petropavlovskikh, P. Effertz and Gary Morris (2023), Challenges of detecting free tropospheric ozone trends in a sparsely sampled environment, submitted to the TOAR-II Community Special Issue, ACP.

Cooper, O. R., D. D. Parrish, A. Stohl, M. Trainer, P. Nédélec, V. Thouret, J. P. Cammas, S. J. Oltmans, B. J. Johnson, D. Tarasick, T. Leblanc, I. S. McDermid, D. Jaffe, R. Gao, J. Stith, T. Ryerson, K. Aikin, T. Campos, A. Weinheimer and M. A. Avery (2010), Increasing springtime ozone mixing ratios in the free troposphere over western North America, Nature, 463, 344-348, doi:10.1038/nature08708

Gaudel, A., O. R. Cooper, et al. (2018), Tropospheric Ozone Assessment Report: Present-day distribution and trends of tropospheric ozone relevant to climate and global atmospheric chemistry model evaluation, Elem. Sci. Anth., 6(1):39, DOI: https://doi.org/10.1525/elementa.291

Gulev, S.K., P.W. Thorne, J. Ahn, F.J. Dentener, C.M. Domingues, S. Gerland, D. Gong, D.S. Kaufman, H.C. Nnamchi, J. Quaas, J.A. Rivera, S. Sathyendranath, S.L. Smith, B. Trewin, K. von Schuckmann, and R.S. Vose, 2021: Changing State of the Climate System. In Climate Change 2021: The Physical Science Basis. Contribution of Working Group I to the Sixth Assessment Report of the Intergovernmental Panel on Climate Change [Masson-Delmotte, V., P. Zhai, A. Pirani, S.L. Connors, C. Péan, S. Berger, N. Caud, Y. Chen, L. Goldfarb, M.I. Gomis, M. Huang, K. Leitzell, E. Lonnoy, J.B.R. Matthews, T.K. Maycock, T. Waterfield, O. Yelekçi, R. Yu, and B. Zhou (eds.)]. Cambridge University Press, Cambridge, United Kingdom and New York, NY, USA, pp. 287–422, doi:10.1017/9781009157896.004

Logan, J. A.: An analysis of ozonesonde data for the troposphere: Recommendations for testing 3-D models and development of a gridded climatology for tropospheric ozone, Journal of Geophysical Research: Atmospheres, 104, 16 115–16 149, 1999.

Pope, R. J., Kerridge, B. J., Siddans, R., Latter, B. G., Chipperfield, M. P., Feng, W., Pimlott, M. A., Dhomse, S. S., Retscher, C., and Rigby, R.: Investigation of spatial and temporal variability in lower tropospheric ozone from RAL Space UV–Vis satellite products, Atmos. Chem. Phys., 23, 14933–14947, https://doi.org/10.5194/acp-23-14933-2023, 2023

Prinn, R.G., 1988. Toward an improved global network for determination of tropospheric ozone climatology and trends. Journal of atmospheric chemistry, 6, pp.281-298.

Saunois, M., Emmons, L., Lamarque, J.-F., Tilmes, S., Wespes, C., Thouret, V., and Schultz, M.: Impact of sampling frequency in the analysis of tropospheric ozone observations, Atmospheric Chemistry and Physics, 12, 6757–6773, https://doi.org/10.5194/acp-12-6757-2012, 2012.

Steinbrecht, Wolfgang, Dagmar Kubistin, Christian Plass-Dülmer, Jonathan Davies, David W. Tarasick, Peter von der Gathen, Holger Deckelmann, Nis Jepsen, Rigel Kivi, Norrie Lyall, Matthias Palm, Justus Notholt, Bogumil Kois, Peter Oelsner, Marc Allaart, Ankie Piters, Michael Gill, Roeland Van Malderen, Andy W. Delcloo, Ralf Sussmann, Emmanuel Mahieu, Christian Servais, Gonzague Romanens, Rene Stübi, Gerard Ancellet, Sophie Godin-Beekmann, Shoma Yamanouchi, Kimberly Strong, Bryan Johnson, Patrick Cullis, Irina Petropavlovskikh, James W. Hannigan, Jose-Luis Hernandez, Ana Diaz Rodriguez, Tatsumi Nakano, Fernando Chouza, Thierry Leblanc, Carlos Torres, Omaira Garcia, Amelie N. Röhling, Matthias Schneider, Thomas Blumenstock, Matt Tully, Clare Paton-Walsh, Nicholas Jones, Richard Querel, Susan Strahan, Ryan M. Stauffer, Anne M. Thompson, Antje Inness, Richard Engelen, Kai-Lan Chang, Owen R. Cooper (2021), COVID-19 Crisis Reduces Free Tropospheric Ozone Across the Northern Hemisphere, Geophysical Research Letters, 48, e2020GL091987. https://doi.org/10.1029/2020GL091987

Szopa, S., V. Naik, B. Adhikary, P. Artaxo, T. Berntsen, W.D. Collins, S. Fuzzi, L. Gallardo, A. Kiendler-Scharr, Z. Klimont, H. Liao, N. Unger, and P. Zanis, 2021: Short-Lived Climate Forcers. In Climate Change 2021: The Physical Science Basis. Contribution of Working Group I to the Sixth Assessment Report of the Intergovernmental Panel on Climate Change [Masson-Delmotte, V., P. Zhai, A. Pirani, S.L. Connors, C. Péan, S. Berger, N. Caud, Y. Chen, L. Goldfarb, M.I. Gomis, M. Huang, K. Leitzell, E. Lonnoy, J.B.R. Matthews, T.K. Maycock, T. Waterfield, O. Yelekçi, R. Yu, and B. Zhou (eds.)]. Cambridge University Press, Cambridge, United Kingdom and New York, NY, USA, pp. 817–922, doi:10.1017/9781009157896.008

Tarasick, DW, et al. 2010. High-resolution tropospheric ozone fields for INTEX and ARCTAS from IONS ozonesondes. J. Geophys. Res. 115(D20): 301. DOI:https://doi.org/10.1029/2009JD012918

Tarasick, D. W., I. E. Galbally, O. R. Cooper, M. G. Schultz, G. Ancellet, T. Leblanc, T. J. Wallington, J. Ziemke, X. Liu, M. Steinbacher, J. Staehelin, C. Vigouroux, J. W. Hannigan, O. García, G. Foret, P. Zanis, E. Weatherhead, I. Petropavlovskikh, H. Worden, M. Osman, J. Liu, K.-L. Chang, A. Gaudel, M. Lin, M. Granados-Muñoz, A. M. Thompson, S. J. Oltmans, J. Cuesta, G. Dufour, V. Thouret, B. Hassler, T. Trickl and J. L. Neu (2019), Tropospheric Ozone Assessment Report: Tropospheric ozone from 1877 to 2016, observed levels, trends and uncertainties. Elem Sci Anth, 7(1), DOI: http://doi.org/10.1525/elementa.376

Wasserstein, R. L., Schirm, A. L., and Lazar, N. A.: Moving to a world beyond $p < 0:05$, Am. Stat., 73, 1–29, https://doi.org/10.1080/00031305.2019.1583913, 2019.

---

## Author Response (AR1)

**Reply on CC1**

The topic of merging ozone data records is of great interest and is indeed a great method to improve sampling and reduce trend uncertainties. The scope of this paper however is mostly to look at the biases and drifts between 2 measurements directly and less so about the individual trends. However, co-author Richard Querel is planning on performing an analysis exactly on this topic of merging the ground-based ozone measurements available at Lauder. This could indeed help resolve discrepancies found with in Pope et al. (2023).

Here, we chose to include a comment on the benefit of using the merged product in future trend studies in the conclusions:

"The good agreement of the three measurements in the troposphere (concerning no significant bias or drift) show that these are reliable to use for trend studies within HEGIFTOM. Future studies can take advantage of this by merging the FTIR, Umkehr and ozonesonde measurements in order to provide more accurate trends thanks to the higher sampling (Chang et al., 2024). However, because no strong correlation is found between Umkehr and FTIR in the troposphere and the Umkehr DOFS or consistently low here, one has to be careful in the inclusion of the Umkehr tropospheric column into the merged product."

Information has been added according to the TOAR-II recommendations:

-Drifts are still shown in %/decade, but now a figure is added (Figure 6) showing those same values in the averaged mole fraction per decade.

-In the abstract it has been specified that it concerns 21$^{st}$ century trends

-A completely new statistical analysis is believed to be out of the scope for the paper in its current state. In section 4.2 (and briefly in the abstract), we have added a statement referring to the language used, especially the term 'statistical significance' such as it is used through the rest of the paper afterwards.

Additional comments have all been accounted for in the manuscript.

**Reply on RC1**

Thank you for all your comments and feedback on the manuscript. We have implemented all your comments in the paper, or when applicable explain here our response. The comments from both referees resulted in significant changes to improve the text accordingly. Most significant changes have occurred in: the introduction to provide additional motivation for the study; the discussion of the results, which have been expanded to provide more scientific explanations; the conclusions to clearly explain the impact of the results on TOAR-II and

LOTUS and the implications for the use of all included instruments; a new appendix per suggestion of an editor comment; and lastly the abstract has been updated to match all the changes. Here I give a short explanation to each of your comments:

- *Table 1 is not clear about the total ozone column measurements (TCO). TCO is provided by FTIR, Umkehr, Dobson and UV2. This does not appear in the table. The table should include a part dedicated to TCO measurements and another one to ozone profiles measurements. In the latter case, it should also include the altitude range of the ozone profiles measurements. Such an information is lacking in section 2 describing the various ground-based instruments.*

The table has been rearranged and a column is added to show which techniques provide total column measurements. Additionally, the information of the vertical extent has been mentioned now explicitly in the sections detailing each measurement technique.

- *References to the ozone time series obtained specifically at Lauder is lacking. For instance, the reference for the intercamparison campaigns of ozone profilers should be McDermid et al., 1998 https://doi.org/10.1029/98JD02706 and a reference to the RIVM ozone lidar instrument should be included, e.g. Swart, Daan P. J., et al., RIVM's Stratospheric Ozone Lidar for NDSC Station Lauder: System Description and First Results,17th International Laser Radar Conference, Sendai, Japan, 405-408, 1994.*

Both these intercomparison studies are now included in the introduction where they are mentioned as past intercomparisons of ozone profilers performed at Lauder. We have added the sentence: "Our study continues intercomparison studies performed at Lauder before 2000 such as by McDermid et al (1998) who look at several ozone profilers (lidar, microwave radiometer, and ozonesonde) and Swart et al. (1995) who focus on RIVM (Rijksinstituut voor Volksgezondheid en Milieu) lidar."

- *Lidar ozone profiles are not highly resolved in the upper stratosphere, see Leblanc et al., https://doi.org/10.5194/amt-9-4029-2016, 2016. This should be mentioned in section 2.4*

A comment about the lidar resolution is added to section 2.4 together with the mentioned reference: "The lidar measurements are well resolved in altitude with the resolution standardized within NDACC according to Leblanc et al. (2016). This resolution ranges from a few hundred meter at 10 km to several kilometers in the upper stratosphere at 50 km."

- *Figure 1 shows partial ozone columns and not total ozone columns for the instruments detecting ozone in specific altitude ranges (e.g. ozonesondes, lidar, microwave spectrometer). This is manifest in the range of partial columns shown and the seasonal variation.*

The caption and y-axis label have been changed to clarify that the time series show integrated ozone column. It is also clarified that this shows partial ozone column for ozonesonde, lidar and microwave radiometer and that it shows total column for FTIR, Umkehr, Dobson, and UV2.

- *Partial columns definition and DOFs: the article is quite explicit on DOFS for FTIR measurements but much less for the quantification of Umkehr and MWR DOFS in the altitude layers selected for partial columns evaluation. More information is needed on these DOFS computation.*

The information on the DOFS for both MWR and Umkehr have been added to their sections 2.2 and 2.3 respectively. Additionally, in section 3.1, both instruments now have the DOFS mentioned as calculated for each partial column.

- *Equation 4: How is handled the discontinuity in the ozone profile (upper range of the profile for the sondes; lower and upper range for the lidar) when the smoothing is applied to these measurements?*

We disregard profiles from MWR, lidar, or ozonesonde if they stop in the middle of a partial column such that no discontinuity needs to be accounted for in the intercomparison. This assures us that all measurements from these three instruments fully cover the altitude extend over which the intercomparison occurs. This clarification has also been added to the text in section 3.1.

- *Time coincidence: More information is needed on the number of measurements made per day made by FTIR and MWR measurements. As for FTIR, various MWR measurements call fall within the time window of other observation. In that case the MWR measurements averaged in the same way? Appendix B does not really answer this question.*

Often both FTIR and MWR have multiple measurements made per day and can fall within

the same time window. The coincidences are constructed by looking at each MWR measurement and finding all the FTIR measurements within the defined window to average out and compare to. If both measurement techniques have multiple measurements in the same day that fall within the time window (for example, one or more FTIR measurements are taken within three hours from two different MWR measurements), then the FTIR measurements that fall in this overlap are used for comparison in both observation pairs. This explanation is added to the section 3.3 for clarity.

• *Bias and dispersion analysis: no reference is given for the scaled MAD computation and more specifically for the 1.4826 factor. More traditionally, standard deviation and standard error (e.g. the standard deviation of the mean) are used to evaluate the significance of the bias between 2 times series. The standard error is then compared to the combined uncertainty averaged over the whole record. How does the methodology used here compares to such traditional method?*

This scaling factor makes the MAD representative as a deviation from the median, similarly as the standard deviation is to the average, in the case of a normal distribution (see Rousseeuw & Croux, 1993). We are not dealing with perfect Gaussian distributions, but the factor still creates a reasonable value for the scatter. The scaled MAD is thus similar to using the standard deviation but is more robust in the sense that it will be less affected by outliers. The mentioned reference has been added to the text in section 4.1.

• *The correlation between 2 ozone time series is heavily driven by the seasonal variation. What would be the correlation for the deseasonalized time series?*

To analyze the correlation without influence of the seasonality, we have added to the table the correlation between monthly anomalies of the time series and discuss these also in the results for each instrument comparison for each partial column.

• *Section 4.1.1: it is not clear if the Dobson TCO corresponds to that computed from the Umkehr retrieval or to the Dobson TCO itself.*

The text had a mistake here and it is clarified which results belong to the Umkehr TCO and which to the Dobson TCO.

• *Section 4.1.5: There is no mention of issues linked to ozone diurnal variation for the*

*comparison in the upper stratosphere, see Sauvageat et al. https://doi.org/10.5194/acp-23-7321-2023, 2023. Impact of ozone diurnal variation on comparison results should thus be discussed.*

The largest diurnal variation is found near the stratopause according to Sauvageat et al. (2023). Our definition of upper stratosphere only reaches up to 42 km. We still estimate the diurnal variation, though, by calculating the short-term variability of the MWR measurements in our upper stratospheric column. We find a maximum value of 18% for the variation in a day, and the variation within the 3-hour window that we consider is very small with a mean value of 1.1%. This is now added to the text in Appendix B.

• Equation 13: how the N parameter corresponding to the number of degrees of freedom is computed?

*The degrees of freedom N, used to calculate the drift error is found from the length of the time series in monthly means that is used to calculate the drift. A small clarification has been added to the text in section 4.2.*

• *Section 4.3: Trend results are interesting and explanations on trend differences due to instrumental drift are convincing. However, if trends are computed for coincident measurements only, the authors should indicate the reduced number of observations used to derive the trends for each data record.*

Because indeed the number of observations used in the intercomparison study has an important effect on the results, we have now added the number of observation pairs that are used for each comparison in table 3.

**Reply on RC2**

Thank you for all your comments and feedback on the manuscript. We have implemented all your comments in the paper, or when applicable explain here our response. The comments from both referees resulted in significant changes to improve the text accordingly. Most significant changes have occurred in: the introduction to provide additional motivation for the study; the discussion of the results, which have been expanded to provide more scientific explanations; the conclusions to clearly explain the impact of the results on TOAR-II and LOTUS and the implications for the use of all included instruments; a new appendix per

suggestion of an editor comment; and lastly the abstract has been updated to match all the changes. Here I give a short explanation to each of your comments:

*Summary of Paper*

*There are five ground-based (GB) instruments at Lauder, New Zealand, that have measured total column ozone and/or partial columns throughout the interval 2000 to 2022. They do not appear to all give the same values in total ozone or in various segments: troposphere, lower stratosphere, middle stratosphere, upper stratosphere. Accordingly, computed trends over the 23-year period differ, especially in the lower stratosphere (LS) where LOTUS has concentrated its efforts. A major goal of this paper, as expressed in the Abstract, is to determine why LOTUS trends are not similar among the techniques. The second goal is to determine "quality and relevance" for TOAR II trends, two criteria that are not well-defined.*

*This paper makes comparisons of the ozone amounts systematically within the segments, using FTIR as the primary reference. Of the four independent measurement types considered, three (Lidar, Microwave, Umkehr) all display significant drift relative to FTIR in one or more stratospheric segments; ozonesondes do not (Table C1). Certain discontinuities near the end of the record contribute to these drifts and the divergence of trends (Appendix D). A reprocessing (modified FTIR retrieval) improves some of the drifts. In summary, the paper contains worthy analyses, carefully carried out.*

*However, there are two reasons why the paper is not ready for publication. First, after all the tables and analyses, the paper does not come back to clear answers to guide how past LOTUS results concerning Lauder can be updated. Nor does the paper provide recommendations for TOAR II activities on how to use the findings in trends analyses. For example, should one try to merge the various datasets for tropospheric ozone analysis? Why or why not? If so, how would that be done?*

In this paper we do not yet focus on the merging of the data sets. The main outcome is to see if there are spatial/temporal/instrumental mismatches between the instruments that can cause a different representation of the ozone field properties. The topic of merging measurements is very interesting though, so we have included a statement about how, thanks to an increased temporal sampling, the trend estimates from merged data sets are improved compared to the individual datasets. This is also work carried out by (a) co-author(s) for future publication.

Additionally, more explicit conclusions have been added on how the results help the research of both LOTUS and TOAR-II.

*The paper needs to be re-outlined and clear conclusions on how, if and why each of the 5 datasets can be used in ongoing LOTUS and TOAR II analyses.*

We have made significant changes to the abstract/introduction to provide a clear context of our goals of the paper concerning all the datasets in context of LOTUS and TOAR-II and we provide a systematic discussion on the results and the repercussions of those results on the usability of separate measurement techniques in the partial columns. This also means that we added more explicit discussions of the issues that cause the biases/drifts in some of the measurement techniques in section 4.2.6.

*Second, there are more fundamental questions about the Lauder datasets relevant to LOTUS and TOAR II. Here are several:*

*1. In the TOAR II HEGIFTOM activity, presumably the FTIR, Umkehr, sonde records have been homogenized. The paper gives no information about the data version, archive, etc, for each of these data sets. Are these the HEGIFTOM files at the RMI ftp repository? The customary doi information is lacking*

Information has been provided in the 'Data availability' section for each of the instruments. A doi is provided for the collection of all data sets used in this article (DOI is currently pending but will soon be available.)

*2. With respect to the ozonesonde data in particular, papers by Stauffer et al (2020; 2022) and updates (through 2021, see Figures below) find total ozone column and stratospheric ozone in particular, suffered the "Ensci dropoff" artifact at Lauder. The upper figure is a satellite comparison – Aura MLS for stratosphere, OMI, OMPS and European TCO comparisons. The lower Figure is based on the Lauder Dobson as archived at WOUDC, Dobson presumably the source of the Umkehr data. Have ozonesonde dropoffs been corrected in the HEGIFTOM files? The wording about the version of sonde data (page 9 of the manuscript) is vague. Reprocessing via the Smit method, even the WMO/GAW, 2021, Report, as referenced (line 217 to 220) do not give a procedure for correcting for the dropoff. Was the process of Nakano & Fujimora, AMT, 2023) to correct the dropoff applied to the Lauder record? "Claim to be removed" is your wording – what does that mean? If the dropoff has been fixed, it would be good to have a supplementary figure showing that.*

In the papers by Stauffer et al. (2020; 2022), Lauder has never been identified as a drop-off site (see Fig. 2 in Stauffer et al., 2020 & Fig. 1 in Stauffer et al., 2022). In Table 2 of Stauffer et al., 2022, the average Lauder EnSci ozonesonde TCO change relative to OMI pre-EnSci and post-EnSci S/N 25,250 is -2.6%, which is below the -3% threshold (defined in Stauffer et al., 2020) for an "Ensci dropoff" site. The overall average of this metric over the entire EnSci network is -1.8%.

Thank you for providing those figures for the Lauder time series. An update of this figure, together with the comparisons with the unhomogenized Lauder ozonesonde time series was presented at the WMO Technical Conference on Meteorological and Environmental

Instruments and Methods of Observation (TECO-2022) in Paris on 10-13 October 2022 and can be downloaded here:

https://ozone.meteo.be/uploads/media/634ea9e9d6f09/vanmalderen-p102-wmoteco2022.pdf?v20231106-1421. From this figure, one could also argue that there is an overall decline in the total ozone content of the ozonesondes w.r.t. the co-located or satellite overpass total ozone measurements, instead of a sudden dropoff. Moreover, Stauffer et al. (2020) also mentioned that "Some sites (e.g., Lauder in 2015) switched radiosondes again from RS-92 to the RS-41", which might have an impact on the ozone profile calculation (through the pump temperature and pressure measurements) as well. Based on these arguments, we do not refer to Lauder as an Ensci dropoff site in the paper.

Surely, Nakano & Fujimora (2023), reported differences in the pump motor specifications of the ozonesondes delivered to the JMA before 2013 (serial numbers ≤24000) and after 2013 (serial numbers >24000), which might have an impact on the total ozone of around 1% (their Fig. 17). Those "measured" JMA pump efficiency correction tables, for each serial number series, might be used instead of the Komhyr et al. (1995) empirical pump correction factors, but there are three problems with this:

• according to the WMO-GAW #268 guidelines, the official reference document for ozonesonde data processing, the Komhyr et al. (1995) tables should be used for En-Sci ozonesondes

• just using the Nakano & Fujimora (2023) pump efficiency measurements instead of the Komhyr et al. (1995) tables will, for SST0.5 and SST1.0 solutions, completely alter the ozone distribution in the upper parts of the profile (starting for pressures lower than 100 hPa) and, as a consequence, the total ozone content of the profile, because the Komhyr et al. (1995) "empirical correction" tables combine decreasing pump efficiency, increasing conversion efficiency, and typical memory effects in the background current for the standard buffered solutions SST1.0 and SST0.5 (Tarasick et al., 2021).

• To solve the incompatibility of the WMO-GAW #268 guidelines with the use of the Nakano & Fujimora (2023) pump efficiency correction tables, the new methodology as decribed in Smit et al. (2024) and in Vömel et al. (2020) is indeed a possibility. But these procedures require some accurate and additional pre-launch information (IB0, IB1, sensor (fast) response time, time between IB1 measurement and launch) and might introduce noise in the data. Practical guidelines to implement those methods under less controlled and well-established conditions as in the JOSIE campaigns, on which data the methods have been developed, are still lacking. As such, these methods are still in research mode and should be implemented and assessed at a global scale before being used in an intercomparison paper like this. Finally, the calibration functions introduced in Smit et al. (2024) to refer the ozonesonde measurements to the photometer in the Jülich simulation chamber, have been determined using the new absorption cross sections for the photometer. To have a consistent comparison between those corrected ozonesonde time record and other co-located techniques, this new set of absorption cross sections should also be used for the other techniques.

To summarize: we will not change the processing of the Lauder ozonesonde time series. It has been processed through its entire time series according to the WMO-GAW #268 guidelines, the official document. The ozonesonde time series at the HEGIFTOM ftp-server have been processed according to those guidelines for all sites, creating a consistency among

those sites at the HEGIFTOM ftp-server. The WMO-GAW #268 does not correct for a possible Ensci dropoff, which has never been identified as such in the Lauder time series. At the moment, the Ensci dropoff has not been corrected for in any site, even in the so-called dropoff sites. New proposed methodologies (Smit et al., 2024 & Vömel et al. 2020) are still in experimental phase and should not be widely used before assessed globally (and after practical guidelines).

We included in the manuscript the sentence: "In a worldwide ozonesonde comparison with satellite and ground-based total column ozone and with satellite stratospheric O3 profiles, Stauffer et al. (2022) mentioned a negative ozone bias in the homogenized Lauder ozonesonde time records in more recent years (see also the plot in https://acd-ext.gsfc.nasa.gov/anonftp/acd/shadoz/nletter/stations_vs_satellites_timeseries.zip). This feature might be related to the so-called "post-2013 dropoff in total ozone" identified in a number of ozonesonde stations (not Lauder) in Stauffer et al. (2020), but, as no clear cause has been determined yet, no correction strategy has been implemented in the here applied WMO/GAW 2021 homogenization procedures."

However, in this response, we want to show the impact of the new ozonesonde processing strategy according to Smit et al. (2024) on the comparison with the FTIR. We have added this in the discussion 4.2.6. where we mention the following about our tests with the new processing:

"While it is decided in the present study to use the sondes data sets from HEGIFTOM that follow the WMO/GAW 2021 homogenization procedures (See Section 2.5), we have performed as a test the drift study on a sonde data set in which a "dropoff" correction as suggested in Stauffer et al (2022) has been applied. The bias and dispersion with FTIR are worsening with this newly processed sonde data set. In the middle stratosphere, where the effect of the dropoff is usually most significant, we see a bias of -9.3% and a scatter of 4.3%. However, it should be mentioned that the drift with FTIR is significantly positive (1.3±1.1 %/decade). This effect is very small in the case of Lauder (1.2 %/decade), but does seem to go in the good direction towards the other ozone stratospheric trend measurements at Lauder where we see a similar positive drift of the measurements with respect to FTIR. To confirm the changing trend when applying the dropoff correction, we perform a similar drift analysis of the ozonesonde data sets to the lidar measurements. In the middle stratosphere we see that the drift (when using (lidar-sonde)/sonde) changes from 2.0±1.3 %/decade for the original ozonesonde data set to 0.7±1.6 %/decade for the newly processed data set. This seems to be consistent with the earlier results comparing sonde to FTIR, because the significant drift between lidar and sonde that is present with the original sonde data is not there for the newly processed sonde data, putting the trends of the ozonesondes more in line with that of lidar.

However, since this newly processed data set is only a temporary test to analyze differences with the original data set, we will remark here that while the EnSci dropoff seems to be better resolved, future attention is needed concerning the ozonesonde trends when a new official ozonesonde data set is available."

*3. If the dropoff has not been corrected, the authors need to implement the Nakano and Fujimora (2023) procedures; ideally the new reprocessing by Smit et al (AMT, 2023) would lead to an even more accurate, referenced result. For LOTUS applications the FTIR-*

*referenced comparisons make sense but for the TOAR II application in the troposphere, the optimized sonde data should also be used as the reference.*

As explained in the response to your previous comment, we stick to the official WMO-GAW #268 processing of the ozonesonde time series. However we do mention in the paper the effect of the new processing on the bias and drift with respect to FTIR and lidar.

*4.    In the case of TOAR II/HEGIFTOM, calculations for 2000-2022 trends being prepared for publication (VanMalderen et al) show the following. Note that trends for the HEGIFTOM ozonesonde data at Lauder (surface to 300hPa) and trends for Umkehr and FTIR at Lauder diverge somewhat as shown below. Graphs of this information were presented to the HEGIFTOM Teams meeting of 7 March. (Based on calculations from NOAA and GSFC)*

*2000-2022 Trends    Surface to 300 hPa    Not rounded to sig fig        QR L1 (ppbv/dec)    QRL3 (ppbv/dec)         MLR L3 (ppbv/dec)*

*Lauder    O3S    -45    169.68    0.134324342    0.01106383        0.133214349*

*   FTIR    -45.04    169.68    1.544135587    1.638209739        1.673699546*

*   Umkehr    -45.04    169.68    0.358046    0.377753        0.579331805*

*It is assumed that the data used in the above Table are the same as Björklund et al are using but more details are required in Section 2.*

The FTIR data used in our study uses a new retrieval method which is found to affect the resulting trend (as can be seen in Figure 7). The new trend of FTIR is lower than with the 'old' FTIR, which then helps to resolve the diverging trends in the troposphere at Lauder.

*RELATED COMMENT IN RESPONSE TO OWEN COOPER COMMENT ON THIS PAPER.(see https://doi.org/10.5194/egusphere-2023-2668-CC1). The table above shows that there is sufficient variation in the surface to 300 hPa trends for sonde, Umkehr and FTIR that "averaging the data" (as Cooper recommends) or averaging the trends is not justified. The current manuscript and the trends analyses show that, in a revised manuscript, more analyses need to be carried out, with careful uncertainty comparisons, on the FTIR, Umkehr and sondes before merging of data can be considered, as suggested by Cooper. It is particularly important that uncertainties for the 5 different instruments being considered are compared.  Note that Figure 1 in the manuscript suggests that FTIR and sonde TCO had some declines, albeit not montonic or identical, after 2014.*

*A further comment on the Cooper et al Comment on this paper. Reference is made to the Pope et al RAL paper: Atmos. Chem. Phys., 23, 14933–14947, 2023*

*https://doi.org/10.5194/acp-23-14933-2023. That paper was accepted prior to the reprocessing of OMI (2014-2021) data that displayed a drift artifact in total ozone. The latter issue is discussed in with corrected data by co-author Ziemke in Gaudel et al: https://egusphere.copernicus.org/preprints/2024/egusphere-2023-3095/. The Pope et al., RAL product overestimates tropospheric ozone trends.*

As mentioned in a reply above, we decided not yet to focus on merging the data sets in this study. The comments of Cooper et al are nonetheless interesting to explore in order to provide a potential way to improve the uncertainties on trend calculations. This is why we chose to include a short discussion on the topic of merging in the conclusions, however also remarking the care that is needed to merge datasets, especially if there is evidence of large dispersion or low correlation between the involved data sets.

*In summary, the paper in its present form should not be published. In a revision the authors need to:*

*•   clarify the source of their data – the customary DOIs and references on the datasets are absent.*

The explanation on the sources in the 'Data availability' section have been expanded and a DOI for the collective dataset will be available for reference.

*•   If the sonde data are not corrected for an artifact stratospheric ozone loss after 2014, that needs to be done before re-analyzing drifts. Intrinsically, the sonde data are more accurate than FTIR in the troposphere and possibly in the lowest and mid-stratosphere. Drifts in FTIR for those segments relative to corrected sonde data should be carried out and discussed for the troposphere, lower and mid-stratosphere.*

Aside from our explanation on the issue of the dropoff in the reply above, we have included an analysis and discussion when comparing the corrected ozonesonde data to both FTIR and lidar in the drift discussion in the results.

*•   Most important, please think through and describe clearly the significance of the new*

*results for LOTUS and TOAR II/HEGIFTOM. The paper currently presents interesting technical details but does not relate a clear scientific story of interest to the TOAR II community.*

Per your suggestion, we have significantly changed the introduction, discussion of results and conclusions to present clearer motivations related to TOAR-II and LOTUS and to provide more concrete and clearer scientific explanations to our results combined with more explicit suggestion of the use of the ground-based measurements in future studies.

*Lesser comments:*

*• Section 2.5. Note that the sonde instrument type and solution used at Lauder should be added.*

We changed this to "At Lauder, ozonesondes from the two different ECC ozonesonde manufacturers (SPC and EnSci, switch made in May 1994) have been launched, and different sensing solution types (SST1.0 and SST0.5, changed in August 1996) been used as well." We also added a reference to the manuscript "Analysis of a newly homogenised ozonesonde dataset from Lauder, New Zealand" by Zeng et al., 2023 in this section: "More details of the Lauder ozonesonde time series and the homogenization procedure can be found in Fig.1, Table 1, and Appendix A of Zeng et al., (2023)."

*On line 214, end of sentence, the following reference for the variations in types of instrument and solutions should be inserted.*

*1. G. J. Smit, A. M. Thompson and ASOPOS, Ozonesonde Measurement Principles and Best Operational Practices, ASOPOS (Assessment of Standard Operating Procedures for Ozonesondes) 2.0, 165 pp., WMO/GAW/IO3C/NDACC/GRUAN, WMO/GAW Report 268, Geneva. (Online at https://library.wmo.int/index.php?lvl=notice_display&id=21986#.YaFNSbpOlc8).*

*Alternatively this can be called WMO/GAW 2021 but the citation is missing from the Reference list at the end of the manuscript*

Thank you for this suggestion. We implemented it.

**Reply on RC2**

Thank you for your comments. Calibration histories are indeed highly relevant in the discussion of an intercomparison of ground-based measurement techniques. Therefore, we have added information about the calibration of the instruments with a focus on FTIR and Dobson instruments, which is added in a fully new appendix (Appendix D).

Concerning the technical comments:

*Page 1, Line 5: Abstract, between => among. Please check usage of "between" here in the abstract and in the manuscript body.*

This change has been made in the manuscript.

*Page 5, Line 109: Technically speaking, this does not show the sensitivity of the instrument itself, but rather of the chosen retrieval strategy.*

This has been changed to refer to the retrieval and not the measurement itself

*Page 23, Line 559: "... thanks new regularization ..." => "... thanks to new regularization ..."*

This change has been made in the manuscript.

*Page 24, Line 578: "... microwindows are chosen ..." => "... microwindows, which are are chosen ..."*

This change has been made in the manuscript.

*Page 25, Line 597: where is this -5.7% shown or derived from?*

The value is derived exactly as in the main body of the paper for the updated FTIR data set. The value here is only mentioned in the text and not in any table.

*Page 26, Line 625- 626: "and we even find that there is complete agreement of FTIR with ozonesonde over all partial columns." This sentence is rather vague.*

This has been changed to "… find that there is no significant drift between FTIR and ozonesonde for any of the partial columns"

*I think the authors should compare and contrast the results with the results of this publication as well:*

*Steinbrecht et al., An update on ozone profile trends for the period 2000 to 2016, Atmos. Chem. Phys., 17, 10675–10690, 2017 https://doi.org/10.5194/acp-17-10675-2017.*

A discussion of their results has been added to the introduction in relation to the similar results found in Godin-Beekmann et al. 2022.

---

## Referee Report (RR1)

Review of Björklund, Vigouroux, Effertz et al: Intercomparison of long-term ground-based measurements of tropospheric and stratospheric ozone at Lauder, New Zealand (45S)

15 July 2024

**Comments on the Revised Paper**

Overall, the paper is much improved and not far from suitable for publication. The most important improvements are: (1) clarifying the data version/quality and the inclusion of the Zeng et al. (2024) reference, that is now available to the reader; (2) the importance of the stratospheric results for LOTUS and stratospheric ozone assessment, ie the rationale for the paper, is better expressed and the theme carried through. Downplaying the TOAR implications, as the authors point out, is appropriate at this point.

The arguments for not reprocessing the sonde data for the "EnSci drop" and related concerns are reasonable *at this time*. However, the ~3% post-2016 total column ozone (TCO) dropoff over Lauder, although seemingly small, remains a robust signal that is affecting the results. The figure below is an update from this Reviewer's prior review, showing the dropoff now in comparison with 5 (rather than 4) TCO satellites. The dropoff in TCO is quite visible in **Figure 1** of the paper and must be stated explicitly by the authors. The dropoff appears to propagate into both the drift (lower stratospheric drift, **Table 4**) and trends (**Table 5**). Indeed, if a 'simple fix' were to add 3% to the post-2016 Lauder sonde data, the **Table 5** trends might decline from ~5%/decade to ~-2%/decade. An important "bottom line" then is that the sonde trend is even smaller than the LOTUS (Godin-Beekmann ref) estimate. Incidentally, there seems to be a stray line in line 623 ? in the manuscript. Line 625, "sonde dataset" not "sondes."

Small edits including the points made in the prior paragraph are needed for an acceptable revision. THANK YOU.

Figure showing ozonesonde 'dropoff' for TCO and stratospheric ozone in the Lauder record (Stauffer et al, 2022 update). Files were downloaded from RMI ftp site, 2024.

---

## Author Response (AR2)

**Reply to referee #1**

Thank you for the additional comments and the noting the erroneous reference in the text. Both remarks have been accounted for in the text.

*This is my second review of the manuscript. The authors have correctly taken into account my remarks and I recommendation publication in Atmospheric Measurement Techniques.*
*I just have 2 minor comments:*

*- line 66: The reference to the ozone lidar implemented in Lauder by RIVM is not correct. It should be:*
*Swart, Daan P. J., et al., RIVM's Stratospheric Ozone Lidar for NDSC Station Lauder: System Description and First Results,17th International Laser Radar Conference, Sendai, Japan, 405-408, 1994*
*This conference paper describes the lidar system and its first results. It is not a comparison paper, so the sentence mentioning the reference to Swart et al., 1994 should be modified accordingly.*

The reference has been corrected and is now moved to the section 2.4 where it fits better to first mention this description of the lidar and its results.

*Line 310: the lidar is also a remote sounding instrument, so I suggest to add the word "passive" in the sentence: The passive remote sounding instruments (FTIR, Umkehr, MWR)*

This is a good suggestion and the word has been added to the text.

**Reply to referee #2**

Thank you for the additional comments. Text has been altered and added to account for your remarks.

*Overall, the paper is much improved and not far from suitable for publication. The most important improvements are: (1) clarifying the data version/quality and the inclusion of the Zeng et al. (2024) reference, that is now available to the reader; (2) the importance of the stratospheric results for LOTUS and stratospheric ozone assessment, ie the rationale for the paper, is better expressed and the theme carried through. Downplaying the TOAR implications, as the authors point out, is appropriate at this point.*

*The arguments for not reprocessing the sonde data for the "EnSci drop" and related concerns are reasonable at this time. However, the ~3% post-2016 total column ozone (TCO) dropoff over Lauder, although seemingly small, remains a robust signal that is affecting the results. The figure below is an update from this Reviewer's prior review, showing the dropoff now in comparison with 5 (rather than 4) TCO satellites. The dropoff in TCO is quite visible in Figure 1 of the paper and must be stated explicitly by the authors. The dropoff appears to propagate into both the drift (lower stratospheric drift, Table 4) and trends (Table 5). Indeed, if a 'simple fix' were to add 3% to the post-2016 Lauder sonde data, the Table 5 trends might decline from ~5%/decade to ~-2%/decade. An important "bottom line" then is that the sonde trend is even smaller than the LOTUS (Godin-Beekmann ref) estimate.*

We added a remark to Section 2.5 concerning the presence of the post-2016 dropoff of ozone with additionally a reference to the supplied figure of the reviewer (although in the previous version with a comparison to 5 satellites instead of 4) where the dropoff is visible.
In Section 4.2.6 two sentences are added to show how the reviewer's suggested 'simple' fix of the dropoff would alter the drift values bringing it closer to the drift values of lidar.
This same change is addressed for its influence on the ozonesonde trends and how they are changed even more compared to the LOTUS trend estimates.
Lastly, this addition to analyze the dropoff is briefly mentioned in the conclusions.

*Incidentally, there seems to be a stray line in line 623 ? in the manuscript. Line 625, "sonde dataset" not "sondes."*

The text has edited to correct for these mistakes.

*Small edits including the points made in the prior paragraph are needed for an acceptable revision. THANK YOU.*